# Efficient Adaptive Filtering for Deformable Image registration

## Abstract

In medical image registration, where targets exhibit piecewise smooth structures, a carefully designed low-resolution data structure can effectively approximate full-resolution deformation fields with minimal accuracy loss. Although this physical prior has proven effective in traditional registration algorithms, it remains underexplored in current learning-based registration literature. In this paper, we propose AdaWarp, a novel neural network module that leverages this prior for efficient and accurate medical image registration. AdaWarp comprises an encoder, a guidance map generator, and a differentiable bilateral grid, enabling an edge-preserving low-frequency approximation of the deformation field. This design reduces computational complexity with low-resolution feature maps while increasing the effective receptive field, achieving a balanced trade-off between registration accuracy and efficiency. Experiments on two registration datasets covering different modalities and input constraints demonstrate that AdaWarp outperforms existing methods in accuracy-efficiency and accuracy-smoothness tradeoffs.

## 1 Introduction

Image registration is a fundamental step in various medical imaging tasks, such as population modeling and statistical atlas construction [1, 2]. Traditional methods [3, 4, 5] minimize an energy function via gradient descent or discrete optimization, often requiring many iterations and extensive hyperparameter tuning. Since they optimize each input pair independently, these methods cannot perform amortized optimization, making it challenging to integrate label supervision from a cohort and leading to slow processing times. Recently, learning-based approaches have accelerated registration by pre-training neural networks on image pair cohorts using amortized optimization. VoxelMorph [6], a seminal model in this domain, leverages a convolutional neural network (ConvNet) [7] to predict deformation fields, achieving fast and accurate image registration.

Several follow-up works have explored different strategies to improve registration accuracy [8, 9, 10, 11, 12, 13, 14, 15, 16, 17, 18]. Many of these approaches leverage advanced network architectures, such as transformers [19, 8, 10] and large convolutional kernels [11, 20], achieving modest accuracy gains at a disproportionately higher computational cost. Moreover, to handle datasets with large deformations, cascaded and pyramid structures such as VTN [16] and LapIRN [9] have been employed to improve registration accuracy; however, they often compromise the balance between registration accuracy and deformation smoothness. While several methods [15, 12, 18, 14], such as DeepFLASH [15] and FourierNet [12], have specifically targeted improving the accuracy-efficiency tradeoff, achieving an optimal balance between the two remains a challenge.

Previous studies have shown that incorporating prior knowledge improves the accuracy-efficiency trade-off in image segmentation [21, 22, 23, 24] and image registration [15, 12, 25], motivating the design of our proposed architecture. Combining this insight with our observations from daily MRI and CT scans in cardiac and abdominal regions, we note two consistent patterns: (1) intensity variations within anatomical regions are generally smooth, and (2) distinct boundaries often exist between organs and the background or neighboring organs. For instance, in cardiac imaging, intensity within regions like the right ventricle and myocardium is relatively homogeneous, while clear and well-defined boundaries are formed by intensity differences between these regions (see Fig. 1, columns 1&2). These consistent intra-region smoothness and inter-region boundaries indicate that certain medical images exhibit piece-wise smooth structures, serving as a valuable physical prior for

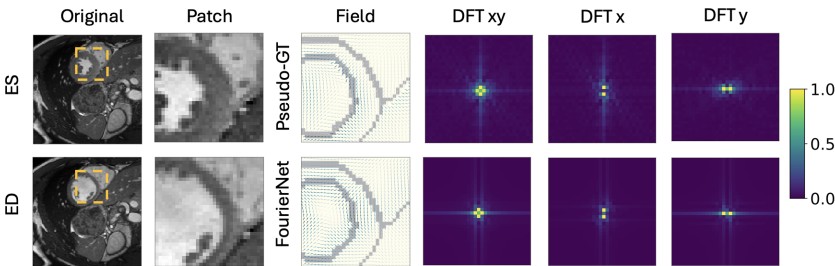

Figure 1: Visual comparison of the deformation fields from end-systole (ES) to end-diastole (ED) generated by both the pseudo-ground truth (GT) and FourierNet [12]. The pseudo-GT was obtained using the DDIR model [31], achieving a Dice (%) exceeding 98% (details in §A) of the appendix. Column 1 displays the original ES and ED images, with Column 2 focusing on the image patches highlighted by the orange box. Column 3 compares the pseudo-GT and FourierNet deformation fields from ES to ED, while Columns 4 to 6 illustrate the corresponding deformation fields in the frequency domain.

image registration. Based on this observation, we propose the following assumption to guide the design of our network module throughout the paper:

**Piece-wise Smooth (P-S) Assumption.** In medical image registration tasks where the targets of interest exhibit piece-wise smooth structures, a carefully designed low-resolution data structure can effectively approximate full-resolution deformation fields.

Several studies [12, 15] have shown that displacement fields exhibit limited high-frequency content in the Fourier domain. This insight allows neural networks to operate in a band-limited space, reducing computational complexity without sacrificing accuracy, particularly in brain MR image registration. Interestingly, follow-up research by Jia et al. [13] demonstrates that FourierNet, leveraging band-limited approximation for cardiac registration, not only reduces computations but also improves accuracy by enlarging the effective receptive field (ERF) [26]. However, reliance on global smoothness constraints makes these methods less effective for datasets with large deformations and complex motions. For instance, as shown in Fig. 1, cardiac imaging involves the heart moving within the rigid thoracic cavity or displaying complex localized motion between ventricles and myocardium, resulting in local discontinuities. In such cases, imposing a globally smooth deformation field becomes too restrictive. To address these discontinuities, some works have employed bilateral filters [27, 28], which preserve edges and improve registration performance in the presence of local discontinuities [29, 30]. However, the non-trainable nature of these filters limits their broader adaptability in learning-based registration frameworks.

Here, we identify a gap in the literature: *existing learning-based registration frameworks lack an end-to-end learnable approach to incorporate the physical prior, i.e., the P-S Assumption, into neural networks, leading to suboptimal registration performance*. In this paper, we address this gap by introducing **AdaWarp**, a neural network module that integrates the P-S assumption, enforcing global smoothness while respecting local discontinuities. AdaWarp employs learnable adaptive filtering to register medical scans, achieving better accuracy-efficiency and accuracy-smoothness trade-offs. AdaWarp comprises an encoder, a guidance mapper, and a differentiable bilateral grid. The encoder, based on ConvNets [32, 33] or Vision Transformers (ViTs) [34], generates low-resolution representations, while the guidance mapper, a multi-layer perceptron (MLP), produces a guidance map representing the range dimension and capturing local intensity differences. The core of **AdaWarp** is the differentiable bilateral grid [35, 36], which inherently incorporates the P-S prior. The grid begins with a differentiable splatting module that maps the 3D image into a 4D bilateral grid, spanning the 3D spatial domain and a 1D range domain. This grid undergoes learnable blurring for adaptive filtering, followed by a slicing module to produce a piece-wise smooth output. The main contributions of this paper are as follows:

- We propose **AdaWarp**, a novel neural network module that integrates the P-S prior in an end-to-end learnable manner, filling the gap of existing learning-based registration frameworks.

- Extensive experiments on two datasets spanning different modalities (MRI & CT) and input constraints (un- and semi-supervised) demonstrate that AdaWarp achieves superior accuracy-efficiency and accuracy-smoothness tradeoffs compared to existing methods.

## 2 RELATED WORK

**Bilateral Grid and High-Dimensional Filtering**: The bilateral filter [27, 28] enhances image quality by replacing each pixel with a weighted average of its neighbors, using weights based on spatial proximity (*spatial domain $\mathcal{S}$*) and intensity similarity (*range domain $\mathcal{R}$*). While effective for edge-preserving image manipulation, its native implementation is slow. Accelerated techniques like the bilateral grid [35, 36], Gaussian KD-Trees [37], permutohedral lattice [38], and adaptive manifolds [39] project signals into compact high-dimensional spaces for real-time performance. These methods have been integrated into neural networks for tasks like scene-dependent image transformation [40] and stereo matching [41], though their use in dense medical image registration remains limited. Recent innovations such as bilateral neural networks [42] and the fast bilateral solver [43] extend these ideas but have yet to find broad applicability in medical imaging.

**Learning with Differentiable Transformations**: The bilateral grid requires both grid-push and grid-pull operations to manage splatting and slicing. While prior studies [44, 41] have represented the range domain using deep bilateral grids and channel shuffling [45], by directly reshaping the encoded tensor $\mathbf{U} \in \mathbb{R}^{C \times H \times W}$ with $C = C_\Gamma \times R$ into a bilateral grid $\mathbf{\Gamma} \in \mathbb{R}^{C_\Gamma \times H \times W \times R}$, this method does not fully capture the range domain. Though conceptually similar, grid-push and grid-pull are adjoint operations. Grid-push handles transformations involving summation, such as Hough transform [46, 47] and splatting in the bilateral grid [35, 36]. Conversely, grid-pull deals with transformations involving spatial sampling, such as image warping using a deformation field [6]. In our work, we use existing differentiable grid-push and grid-pull techniques [48, 49] to build the differentiable grid.

**Learning-based Image Registration**: Image registration aims to align a moving image $\mathbf{I}_m$ with a paired fixed image $\mathbf{I}_f$ by estimating a deformation field $\phi$. The transformation at each voxel is defined as $\phi(x) = x + \mathbf{u}(x)$, where $x$ is a spatial location in the domain $\Omega \subseteq \mathbb{R}^{H \times W \times D}$, and $\mathbf{u}(x)$ is the displacement vector at $x$. This deformation field $\phi$ establishes a voxel-wise mapping from each location in $\mathbf{I}_f$ to its corresponding location in the warped moving image $\mathbf{I}_m \circ \phi$. Learning-based methods, such as VoxelMorph [6], employ unsupervised learning to optimize the expected loss function and derive neural network weights $\theta$ from a cohort of image pairs $D$, formulated as:

$$\hat{\theta} = \arg\min_{\theta}\{\mathbb{E}_{(\mathbf{I}_f,\mathbf{I}_m)\sim D}[f(\mathbf{I}_f, \mathbf{I}_m \circ g_\theta(\mathbf{I}_f, \mathbf{I}_m)) + \lambda r(g_\theta(\mathbf{I}_f, \mathbf{I}_m))]\}. \tag{1}$$

Here, $f$ and $r$ represent the dissimilarity and regularization functions, respectively. The function $g_\theta$, once trained, predicts the deformation field $\phi$ directly from the input, i.e., $\phi = g_\theta(\mathbf{I}_f, \mathbf{I}_m)$. Recent advancements in learning-based image registration extend beyond VoxelMorph [6] by incorporating Vision Transformers [19, 50] into frameworks like TransMorph [8] and XMorpher [10]. Innovations such as symmetric networks [51, 52], multi-channel architectures [31], large-kernel convolutions [11, 53], and cascaded frameworks [54, 9, 17] have further advanced the field. More recent methods like RDP [55] and CorrMLP [56] combine image pyramids [57] or multi-scale strategies [17] with advanced modules like MLPs and vision transformers [58], achieving competitive performance. However, as noted by Jena et al. [59] and Hansen et al. [60], amortized optimization with the same voxel-wise dissimilarity as iterative instance optimization methods offers no clear advantages in unsupervised settings. AdaWarp addresses this gap by integrating the P-S prior, naturally modeling pairwise voxel relationships and enhancing flow signal propagation within a learned cost volume in an edge-aware manner.

## 3 ADAWARP

In this section, we present the details of AdaWarp, starting with an introduction to the traditional bilateral grid. We then describe how AdaWarp incorporates a differentiable bilateral grid into the current backbone network. While AdaWarp is applied to 3D volumetric images using a 4D grid, for simplicity, we illustrate the framework using a 2D spatial domain.

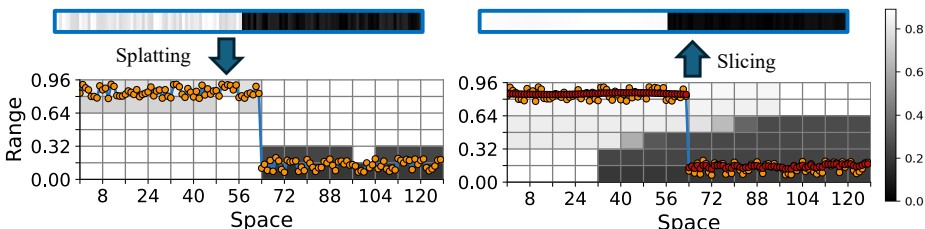

Figure 2: Visual illustration of the bilateral grid process on a random step function. **Splatting**: Projects the 1D step function from a (1x128) space into a 16x6 sparse spatial-range grid using nearest interpolation with sampling rates $s_s = 8$ and $s_r = 0.16$. **Blurring**: Gaussian filters with $\sigma = 1$ to both spatial and range dimensions, corresponding to $\sigma_s = \sigma \times s_s$ and $\sigma_r = \sigma \times s_r$ in the initial image space. **Slicing**: Utilizes linear interpolation for slicing, normalized via homogeneous coordinates. **Left:** The initial step function is shown in yellow dots, with the projected intensities linked by blue lines in the bilateral grid. **Right:** The blurred bilateral grid and the filtered signal are visualized, with red dots representing the sliced output compared to the original yellow signal.

### 3.1 PRELIMINARIES

Here we briefly review the process of implementing fast bilateral filtering via a bilateral grid [35, 36] (ref to Fig. 2 for an visual example). Consider a guidance image $\mathbf{G} \in \mathbb{R}^{h \times w}$ normalized to the range $(0, 1)$, and an input image $\mathbf{I} \in \mathbb{R}^{h \times w}$. Let $s_s$ and $s_r$ denote the sampling rates in the spatial domain $\mathcal{S}$ and the range domain $\mathcal{R}$, respectively. We can establish a bilateral grid $\mathbf{\Gamma} \in \mathbb{R}^{\lceil \frac{h}{s_s} \rceil \times \lceil \frac{w}{s_s} \rceil \times \lceil \frac{1}{s_r} \rceil \times 2}$. This grid is initially set to zero and then updated by accumulating homogeneous coordinates $(\mathbf{I}(x, y), 1)$:

$$\mathbf{\Gamma} \left( \left[ \frac{x}{s_s} \right], \left[ \frac{y}{s_s} \right], \left[ \frac{\mathbf{G}(x, y)}{s_r} \right] \right) \mathrel{+}= (\mathbf{I}(x, y), 1). \tag{2}$$

Here, $(x, y)$ are the original **image coordinates**, and the triplet $(x, y, \mathbf{G}(x, y))$ represents **grid coordinates** for accessing elements in the bilateral grid, with $[\cdot]$ denoting the rounding operation. The process described in Eq. (2) is called **splatting**, where image signals are projected onto the higher-dimensional space $\mathcal{S} \times \mathcal{R}$. Any function $f$, including neural networks that take the constructed grid $\mathbf{\Gamma}$ as input, can blur (manipulate) the grid, producing $\tilde{\mathbf{\Gamma}} = f(\mathbf{\Gamma})$. Subsequently, slicing generates a new image by sampling at grid location $\left( \frac{x}{s_s}, \frac{y}{s_s}, \frac{\mathbf{G}(x, y)}{s_r} \right)$ using multi-linear interpolation. This tri-phase splatting-blurring-slicing process serves as the primary building block of **AdaWarp**.

### 3.2 DIFFERENTIABLE BILATERAL GRID

Adopting notations from prior research [49, 48], we formally define the differentiable splatting and slicing operations in the following sections. While high-dimensional filtering can project signals onto arbitrary spaces, we focus on extending by one additional dimension. This single range dimension is sufficient to respect object boundaries by capturing pairwise voxel intensity differences, thereby implementing the P-S Assumption. Extending to higher dimensions is possible and could generalize adaptive filtering further, but this lies beyond the scope of the current paper. See the appendix for a preliminary derivation.

#### 3.2.1 SPLATTING, BLURRING & SLICING

Splatting and slicing are symmetric operations, essential for producing piece-wise smooth outputs. Given a source tensor $\mathbf{U} \in \mathbb{R}^{H \times W \times C}$ and a target bilateral grid $\mathbf{\Gamma} \in \mathbb{R}^{H' \times W' \times R \times C}$, where $H, W$ are input and $H', W'$ are output spatial dimensions, $R$ is the range dimension size, and $C$ is the number of channels. A sampling grid $\mathbf{G} \in \mathbb{R}^{H \times W \times 3}$, consisting of $\mathbf{G}^x$ and $\mathbf{G}^y$ as mesh grids and $\mathbf{G}^r$ as the guidance map, along with a kernel function $\mathcal{K}()$, are used. The accumulated values at a

Figure 3: Overview of the AdaWarp framework, which consists of an encoder, a guidance map generator implemented with an MLP, and learnable adaptive filtering using a differentiable bilateral grid. Details on the splatting and slicing processes are provided in §3.2, while the overall workflow of the framework is described in §3.3.

cell $\mathbf{\Gamma}_{ijk} \in \mathbb{R}^C$ and the value at cell $(n,m)$ in the sliced feature map $\mathbf{V} \in \mathbb{R}^{H \times W}$ are determined as:

$$\mathbf{\Gamma}_{ijk} = \sum_{(n,m)}^{\mathcal{S}} \mathbf{U}_{nm}\mathcal{K}(\mathbf{G}_{nm}^x, i)\mathcal{K}(\mathbf{G}_{nm}^y, j)\mathcal{K}(\mathbf{G}_{nm}^r, k), \tag{3}$$

$$\mathbf{V}_{nm} = \sum_{(i,j,k)}^{\mathcal{S} \times \mathcal{R}} \tilde{\mathbf{\Gamma}}_{ijk}\mathcal{K}(\mathbf{G}_{nm}^x, i)\mathcal{K}(\mathbf{G}_{nm}^y, j)\mathcal{K}(\mathbf{G}_{nm}^r, k), \tag{4}$$

Here the kernel function $\mathcal{K}()$ can be any predefined kernel, such as the linear interpolation kernel $\mathcal{K}(p,q) = \max(0, 1 - |p - q|)$. Eq. (3) can be represented as $\mathcal{D} : (\mathbf{U}, \mathbf{G}; \mathcal{K}) \mapsto \mathbf{\Gamma}$, and Eq. (4) can be represented as $\mathcal{F} : (\tilde{\mathbf{\Gamma}}, \mathbf{G}; \mathcal{K}) \mapsto \mathbf{V}$. The $\tilde{\mathbf{\Gamma}}_{ijk}$ is generated through a "blurring" function, implemented using a learnable neural network layer composed of two convolutional layers. This transforms the bilateral grid, where $\tilde{\mathbf{\Gamma}} = f(\mathbf{\Gamma})$ represents the blurring operation applied to $\mathbf{\Gamma}$. The difference between splatting and slicing lies in the data flow dynamics. In splatting, each cell in $\mathbf{U}$ "pushes" its values to a specific location in $\mathbf{\Gamma}$. Conversely, in slicing, each cell in $\mathbf{V}$ "pulls" values from a specific location in $\tilde{\mathbf{\Gamma}}$. In the equations, for splatting (Eq. 3), each cell on the left side may accumulate values from multiple locations on the right, whereas in slicing (Eq. 4), each cell on the left side typically samples from a single location on the right. We refer to these as locations rather than cells on the right side because, with the use of a multi-linear kernel, a single location can correspond to multiple cells. For additional details on gradient computations for both processes, see [49, 48].

### 3.2.2 ACHIEVING ADAPTIVE FILTERING

For edge-preserving filtering in the bilateral grid, it is essential to track the pixel count or weight per grid cell. During splatting, a tensor the same spatial size as $\mathbf{U}$, filled with ones, is concatenated with $\mathbf{U}$ across channels. The grid is then divided into a value tensor and a weight tensor post-splatting, with the former normalized by the latter before further processing.

Filtering in the original image maintains translation-equivariance in the spatial domain $\mathcal{S}$ but can propagate information across nearby objects, causing issues in image registration with local discontinuities. In contrast, filtering in the bilateral grid preserves locality in both spatial and range domains $\mathcal{S} \times \mathcal{R}$, enabling edge-preserving filtering. As the same filter is applied to each cell in the grid, originally adjacent cells in the spatial domain may become farther apart in the range domain. When projected back to image space, each pixel is effectively filtered with a unique kernel, adapting to local intensity variations, akin to the adaptability of self-attention mechanisms. See the appendix for the derivation of connections between adaptive filtering and self-attention.

### 3.3 ADAWARP FRAMEWORK

In this section, we introduce the AdaWarp framework, designed to improve the accuracy-efficiency trade-off in image registration by leveraging the piece-wise smooth (P-S) prior, as discussed in

§1. As shown in Fig. 3, AdaWarp consists of an image encoder, a guidance map generator, and a differentiable bilateral grid (detailed in §3.2).

The encoder, which could be any existing backbone network such as ResNets [32] or a U-Net [7] with linear interpolation, reduces the spatial size of the input image $(H_0, W_0)$ by a certain factor, producing a feature map of spatial size $(H, W)$. This feature map serves as a low-resolution approximation of the original input images. The moving and fixed feature maps are used to compute disparities, which are then reshaped to form a cost volume tensor $\mathbf{U} \in \mathbb{R}^{H \times W \times C}$, similar to ConvexAdam [61, 62]. We use a multi-layer perceptron (MLP), following prior work [40, 41], to generate a single-channel guidance map $\mathbf{G}^r$ in the range $(0, 1)$. By applying a sampling rate $s_r$, we compute $\frac{\mathbf{G}^r}{s_r}$ as the grid coordinate for the range domain. This, along with the spatial coordinates $\frac{\mathbf{G}^x}{s_s}$ and $\frac{\mathbf{G}^y}{s_s}$ of the original image, forms the complete sampling grid $(\frac{\mathbf{G}^x}{s_s}, \frac{\mathbf{G}^y}{s_s}, \frac{\mathbf{G}^r}{s_r})$. The sampling grid enables access to elements in the bilateral grid, supporting the splatting and slicing processes. The feature map $\mathbf{U}$ is then splatted onto a bilateral grid $\mathbf{\Gamma} \in \mathbb{R}^{H' \times W' \times R \times C}$ and blurred via learnable convolution layers to yield the refined grid $\tilde{\mathbf{\Gamma}} \in \mathbb{R}^{H' \times W' \times R \times C'}$. Slicing this grid recovers the original spatial size $(H_0, W_0)$ in the final feature map $\mathbf{V} \in \mathbb{R}^{H_0 \times W_0 \times C}$, resulting in spatial and range sampling rates set at $s_s = \frac{H_0}{H'} = \frac{W_0}{W'}$ and $s_r = \frac{1}{R}$, respectively.

## 4 EXPERIMENTS & RESULTS

In this section, we evaluate AdaWarp on two tasks: unsupervised cardiac cine-MR registration and semi-supervised abdomen CT registration. We detail the datasets, implementation, baselines, and metrics, followed by results and analysis focusing on accuracy-efficiency and accuracy-smoothness trade-offs.

### 4.1 DATASETS, IMPLEMENTATION DETAILS, BASELINE METHODS & EVALUATION METRICS

**Cardiac Dataset (Unsupervised Learning).** We evaluate unsupervised intra-subject cardiac cine-MR image registration on the ACDC dataset [63], containing 150 subjects with ED and ES phase images and segmentation masks for the right ventricle (RV), left ventricular myocardium (LVM), and left ventricular blood pool (LVBP). We register ED to ES images and vice versa, resulting in 300 image pairs. The dataset is split into 170 training, 30 validation, and 100 testing pairs, with no subject overlap. All images are normalized to (0,1), resampled to a voxel size of 1.8x1.8x10 mm, and cropped to 128x128x16. In the unsupervised setting, no masks were used for training or testing.

**Abdomen Dataset (Semi-supervised Learning).** We evaluate inter-subject multi-organ registration on the Abdomen CT dataset [64], which includes 30 scans with 13 segmented structures. The dataset was split into 380 pairs (20×19) for training, 6 pairs (3×2) for validation, and 42 pairs (7×6) for testing. All images were resampled to a voxel size of 2 mm, resized to 192×160×256, and min-max normalized to $(0, 1)$ with intensities clipped to $[-800, 500]$ Hounsfield units. In the semi-supervised setting, masks were used only during training.

#### 4.1.1 TRAINING DETAILS AND BASELINE METHODS

All experiments and baseline methods were conducted using Python 3.7 and PyTorch 1.9.0 [65] on an A100 GPU and a 16-core CPU. TorchScript was used to implement splatting and slicing for performance optimization. Training details can be found in §C of the appendix.

**Baseline Methods.** We benchmark **AdaWarp** against leading learning-based models, including VoxelMorph [6], TransMorph [8], LKU-Net [11], FourierNet [12], CorrMLP [56], and RDP [66]. For the cardiac dataset, we also include MemWarp [17], a recently developed multi-scale network specifically for cardiac registration, and DeBG, a deep bilateral grid model previously used in image manipulation [40] and stereo matching [41]. For the abdomen CT dataset, we include LapIRN [9] and textSCF [25], both designed for handling large deformations. Additionally, we evaluate discrete optimization methods ConvexAdam [67] and SAMConvex [68], tailored for dataset with large deformations and limited instances.

| Model | Avg. (%) | RV (%) | LVM (%) | LVBP (%) | HD95 ↓ | SDlogJ ↓ |
|---|---|---|---|---|---|---|
| Initial | 58.14 | 64.50 | 48.33 | 61.60 | 11.95 | - |
| VoxelMorph [6] | 76.35* | 74.69 | 73.19 | 81.15 | 9.28 | 0.049 |
| TransMorph [8] | 76.89* | 75.39 | 73.52 | 81.75 | 9.11 | 0.049 |
| FourierNet [12] | 77.04* | 75.30 | 73.88 | 81.96 | 9.10 | 0.045 |
| LKU-Net [11] | 77.10* | 75.16 | 74.20 | 81.75 | 9.14 | 0.048 |
| MemWarp [17] | 77.25* | 75.86 | 73.92 | 81.99 | 9.23* | 0.074 |
| DeBG [40] | 77.36* | 76.05 | 74.41 | 81.61 | **8.75** | 0.042 |
| CorrMLP [71] | 77.58* | 74.84 | 75.68 | 82.21 | 9.23* | 0.052 |
| RDP [66] | 77.62* | 74.70 | 75.95 | 82.20 | 9.15 | 0.050 |
| **Ada-Cost (Ours)** | **79.82** | **77.58** | **77.95** | **83.92** | 8.98 | 0.050 |

Table 1: Quantitative evaluation of different models on the ACDC dataset. Top scores are highlighted in bold. Metrics include Average Dice (%), RV Dice (%), LVM Dice (%), LVBP Dice (%), HD95 (mm), and SDlogJ, with ↓ indicating lower is better.

| Model | Type | Dice (%) | HD95 ↓ | SDlogJ ↓ |
|---|---|---|---|---|
| Initial | - | 30.86 | 29.77 | - |
| FourierNet [12] | L | 42.80* | 22.95* | 0.13 |
| VoxelMorph [75] | L | 47.05* | 23.08* | 0.13 |
| TransMorph [8] | L | 47.94* | 21.53* | 0.13 |
| LapIRN [70] | L | 51.39* | 20.89* | 0.06 |
| LKUNet [11] | L | 52.08* | 20.34* | 0.28 |
| CorrMLP [71] | L | 56.58* | 20.40* | 0.16 |
| RDP [66] | L | 58.77* | 20.07* | 0.22 |
| TextSCF [25] | L | 60.75* | 22.44* | 0.87 |
| ConvexAdam [67] | D | 51.10* | 23.14* | **0.11** |
| Ada-ConvexAdam | D | 51.29* | 23.29* | **0.11** |
| SAMConvex [68] | D | 53.65* | 18.66* | 0.12 |
| Ada-SAMConvex | D | 53.94* | 18.52* | 0.12 |
| **Ada-Cost (Ours)** | L&D | **64.97** | **13.70** | 0.17 |

Table 2: Quantitative comparison on the Abdomen CT dataset. "L" denotes learning-based methods, and "D" represents discrete optimization-based methods.

### 4.1.2 IMPLEMENTATION DETAILS

For all methods, including **AdaWarp**, we follow Balakrishnan et al. [6] and use *scaling and squaring* [69] with 7 integration steps for diffeomorphic transformation. It is worth noting that the guidance map is sourced from the fixed image only, as for each voxel location in the target image, the deformation field samples a value from moving image. Dataset-specific implementation details are provided below.

**Ada-Cost.** The instantiation of **AdaWarp** for deformable image registration is **Ada-Cost**, which follows ConvexAdam's discrete optimization strategy [62, 61] while being **end-to-end trainable** like other learning-based methods. It uses two 3D conv-norm-act blocks as encoder to extract moving and fixed feature maps, followed by trilinear downsampling to form a 3-level image pyramid. At each pyramid level, a cost volume with one neighbor is computed, followed by two 4D conv-norm-act blocks for adaptive filtering, and finalized with a 3D conv-norm-act block and a convolution to extract the deformation field. Unlike prior multi-scale methods [70, 71, 17], Ada-Cost processes cost volumes with shared network weights across pyramid levels, reducing parameters while maintaining optimal performance. For the cardiac dataset, raw images are processed through an additional conv-norm-act block before being input to the encoder. For the abdomen dataset, feature maps extracted by a pretrained universal segmentation network [72] (pre-softmax) serve as input to the encoder.

**Dataset specifics.** In cardiac dataset, for DeBG, aside from the encoder and splatting adapted from [44, 41], all other components (e.g. spatial and range sampling rates) follow the Ada-Cost setup. In DeBG, FourierNet, and Ada-Cost, downsampling is omitted in the axial direction due to slice thickness considerations. Discrete optimization methods use pretrained feature descriptors: ConvexAdam utilizes MIND [73], while SAMConvex employs contrastively pretrained descriptors from a large CT dataset [74]. As SAMConvex's pretrained model is unavailable, we replace it with a pretrained CT segmentation model [72], the same used as input to Ada-Cost. Both methods adopt a 3-level image pyramid. In Table 2, Ada-* denotes the use of a non-learnable bilateral filter for cost volume filtering with Gaussian kernels ($\sigma = 1$) for both spatial and range domains.

### 4.1.3 EVALUATION METRICS.

Following standard practice [6, 8], we use the Dice Similarity Coefficient (Dice) and 95th percentile Hausdorff Distance (HD95) to assess anatomical alignment, and the standard deviation of the Jacobian determinant's logarithm (SDlogJ) to measure deformation smoothness. Computational complexity is evaluated using multiply-add operations (Multi-Adds, G) and parameter size (Params, MB). Statistical significance is determined using paired t-tests on both Dice (%) and HD95, with an asterisk (*) indicating significance for comparison methods at $p < 0.05$. Absence of an asterisk denotes no significance.

## 4.2 RESULTS AND ANALYSIS

### 4.2.1 QUANTITATIVE RESULTS & ANALYSIS

**Cardiac Dataset.** Table 1 compares methods on the ACDC dataset, highlighting that all approaches produce smooth deformation fields with low SDlogJ. Ada-Cost achieves the highest Dice scores across

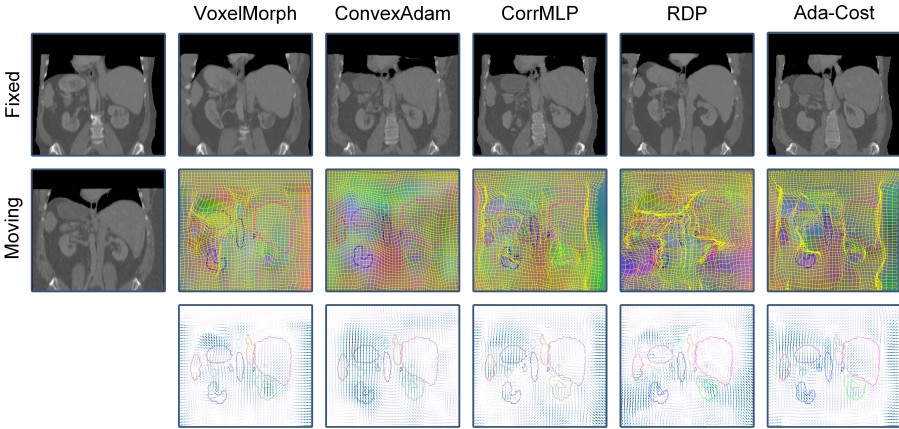

Figure 4: Qualitative results on the abdomen dataset. The first row shows the original fixed image alongside the warped moving images produced by each method. The second row displays the original moving image and the deformation fields in grid format for each method. The third row illustrates the projected 2D vector fields of each method. Color contours highlight objects of interest, with different organs represented in distinct colors.

all anatomical structures, surpassing DeBG, which uses shuffled channels for range representation but lacks Ada-Cost's performance. Multi-scale methods like RDP, CorrMLP, and MemWarp outperform single-scale approaches like VoxelMorph and LKU-Net in Dice (%). Ada-Cost exceeds the runner-up RDP by 2.83% in Dice (%), while maintaining comparable deformation smoothness measured by SDlogJ. The most notable improvement is observed for the right ventricle (RV), likely due to the complex motion between RV and LV compared to the relatively uniform motion within LVM and LVBP. This highlights the importance of preserving local discontinuities and leveraging piece-wise smooth properties rather than relying purely on band-limited approaches. The comparison between Ada-Cost and DeBG further emphasizes the benefits of actual splatting to maintain the image manifold in high-dimensional space, as opposed to reshaping tensors for range representation. Interestingly, while not observed for the abdomen dataset, multi-scale methods like Ada-Cost exhibit slightly higher HD95 compared to their single-scale counterparts.

**Abdomen Dataset.** Table 2 compares various methods on the Abdomen dataset. Discrete optimization methods generally outperform single-scale learning-based approaches, except for TextSCF. While TextSCF achieves high Dice scores by leveraging pretrained segmentation masks with a visual-language model to steer dynamic filters, it produces implausible deformation fields, as indicated by an SDlogJ of 0.87. Multi-scale methods, including CorrMLP, RDP, and Ada-Cost, excel in anatomical alignment accuracy while maintaining relatively smooth deformation fields. Specifically, Ada-Cost, with similar or slightly higher SDlogJ, surpasses the best-performing multi-scale approach RDP and the discrete optimization method SAMConvex by 10.54% and 21.10% in Dice improvement, respectively. In HD95 reduction, Ada-Cost achieves improvements of 31.73% over RDP (20.07 to 13.70) and 26.61% over SAMConvex (18.66 to 13.70). Interestingly, the bilateral filter counterparts of ConvexAdam and SAMConvex slightly outperform their original versions in Dice but fall short compared to Ada-Cost. This highlights the superiority of using learnable adaptive filtering over fixed-kernel bilateral filtering.

### 4.2.2 QUALITATIVE RESULTS & ANALYSIS

**Abdomen Dataset.** The qualitative results of Ada-Cost compared to other baseline methods on the abdomen dataset are shown in Fig. 4. **From the first row**, Ada-Cost achieves the closest anatomical alignment to the fixed image in two notable aspects: (1) For the left and right kidneys, which have large displacements nearly exceeding their size, only Ada-Cost preserves kidney integrity without affecting surrounding tissues (e.g., ConvexAdam misaligns the liver, and other methods distort the vertebral spine); (2) Ada-Cost and CorrMLP are the only methods to correctly capture and respect the body margin near the image boundary visible in the fixed image. **From the second and third rows**, we observe the following: (1) Only Ada-Cost and CorrMLP produce outward-pointing vectors

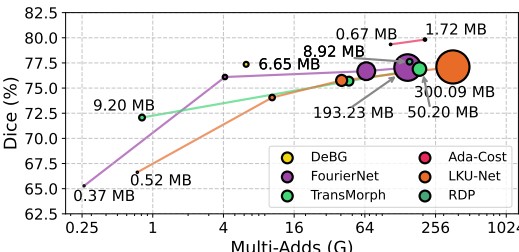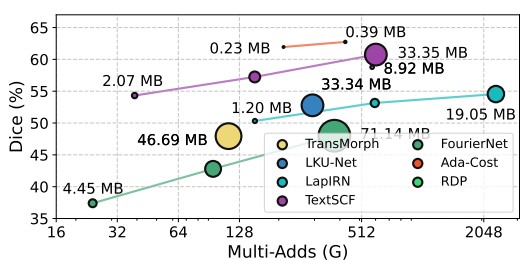

Figure 5: Visual comparison of the trade-off between Avg. Dice (%) and computational complexity on the cardiac dataset, with Multi-Adds (G) on a logarithmic x-axis and Params (MB) indicated by circle size. TransMorph is shown in three versions (tiny, small, normal), while complexity for FourierNet, LKU-Net, RDP, and Ada-Cost is adjusted by varying the initial channel count.

Figure 6: Visual comparison of the trade-off between Avg. Dice (%) and computational complexity on the abdomen dataset, with Multi-Adds (G) on a logarithmic x-axis and Params (MB) indicated by circle size and label. TransMorph is shown in its normal version, while complexity for others and Ada-Cost is adjusted by varying the initial channel count.

at the left/right boundaries, while others point inward, failing to capture the margins. (2) Ada-Cost exhibits certain foldings in the displacement field, but the displacements within each object remain smooth, e.g., only ConvexAdam and Ada-Cost show smooth fields within the left kidney (left refers to the image position throughout). (3) Ada-Cost effectively preserves local discontinuities and sharp object boundaries, such as the left/right kidneys and the right side of the liver, while RDP shows discontinuities inside organs, which is implausible.

**Cardiac Dataset.** Qualitative results for Ada-Cost compared to other methods on the ACDC dataset are shown in Fig. 10 in §F of the appendix. Key observations: (1) Despite similar SDlogJ values, Ada-Cost produces smoother fields, particularly in background regions and transitions between the RV and LV myocardium. (2) When registering end-diastole to end-systolic phases, only Ada-Cost exhibits a single realistic center in the RV with inward-pointing vector fields, while others show two.

### 4.2.3 RESULTS & ANALYSIS ON COMPUTATIONAL COMPLEXITY

We evaluate network complexity using Multi-Adds (G) and Params (MB), adjusted by parameter size for each network. Fig. 5 compares Ada-Cost's accuracy and complexity against other methods on the ACDC dataset. Table 1 highlights Ada-Cost achieving 79.82% Dice with 1.72 MB and 208 G Multi-Adds. A smaller version achieves 79.34% Dice with 0.67 MB and 106 G Multi-Adds, outperforming the runner-up RDP by 2.21% in Dice, reducing parameter size by 92.48%, and Multi-Adds by 31.17%. This demonstrates Ada-Cost's superior accuracy-efficiency trade-off on the cardiac dataset compared to all baselines.

Fig. 6 compares the accuracy-efficiency tradeoff of Ada-Cost with other methods on the abdomen dataset. By adopting discrete optimization and using pretrained feature extraction with shared weights across pyramid levels, Ada-Cost reduces the parameter size to under 1 MB while substantially improving registration accuracy in terms of Dice and HD95. Similar to TextSCF [25], which incorporates external segmentation masks during training and inference, Ada-Cost and SAMConvex [68] utilize pretrained feature maps for enhanced performance. Ada-Cost (0.23 MB) achieves a slightly higher Dice (61.94%) compared to the runner-up TextSCF (60.75%, 33.35 MB), while reducing Multi-Adds by 64.91% and Params by 99.31%. However, it is important to note that the feature extractor itself adds 1110 G Multi-Adds and 19.07 MB in Params. Future work will focus on developing more efficient feature extractors using contrastive learning or masked auto-encoders.

### 4.2.4 RESULTS & ANALYSIS ON DEFORMATION FIELD SMOOTHNESS

We omit smoothness plotting for the cardiac dataset, as all methods produce smooth deformation fields with SDlogJ below 0.1, the largest being 0.074 from MemWarp [17]. Details on the plausibility and regularity of the deformation fields are provided in Fig. F in the appendix.

In contrast, the abdomen dataset exhibits greater variability in smoothness due to inherently larger deformations, which requiring careful handling to prevent overfitting to segmentation masks and the generation of implausible fields. As shown in Fig. 7, Ada-Cost with varying $\lambda : 10.0 \rightarrow 1.0 \rightarrow 0.1 \rightarrow 0.01$ in Eq. (1) shows that Dice scores peak at an SDlogJ of 0.17 before declining, indicating strong anatomical alignment across different smoothness levels. Notably, with similar Dice to TextSCF (60.1% vs. 60.75%), Ada-Cost ($\lambda = 10.0$) achieves a much smaller SDlogJ (0.10 vs. 0.87). FourierNet, while producing smooth deformation fields due to its band-limited design, fails to capture local discontinuities, leading to poor anatomical alignment. This high-

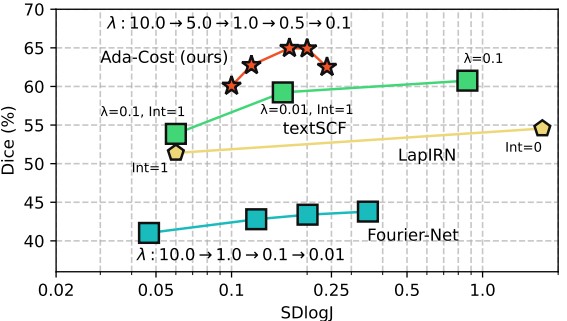

Figure 7: Trade-off between smoothness and Dice score (%) for Ada-Cost and benchmarks. Smoothness regularization $\lambda$ for Ada-Cost is listed on top, while Fourier-Net parameters are below. 'Int' indicates integration was applied.

lights the effectiveness of incorporating the proposed P-S assumption, which is crucial for certain medical image registration tasks.

## 5 DISCUSSIONS

### 5.1 EFFECTS OF SAMPLING RATE $s_r$ & $s_s$.

The sampling rate $s_r$ affects the range domain ($\mathcal{R}$). At $s_r = 1$, the range dimension reduces to 1, turning the process into a **gated attention process**, where the guidance map gates the linearly upsampled encoder output. Decreasing $s_r$ increases range dimensionality, enhancing edge distinction but also raising computational costs. Experiments show that reducing $s_r$ from 1 to $1/64$ improves accuracy in both cardiac and abdomen registration, peaking at $s_r = 1/8$ and $s_r = 1/32$, respectively, before declining. The smaller optimal $s_r$ for abdomen reflects its higher need for handling local discontinuities. The sampling rate $s_s$ influences the spatial domain ($\mathcal{S}$). At $s_s = 1$, spatial dimensions match the original image, and increasing $s_s$ smooths the results while reducing computation. Raising $s_s$ from 1 to 32 improves accuracy, peaking at $s_s = 8$ for both tasks before decreasing.

### 5.2 LIMITATIONS.

The development of AdaWarp revealed two key limitations. **First**, while AdaWarp effectively respects object boundaries by projecting image signals to a higher-dimensional space, we only explored intensity differences for edge awareness. As discussed in §B of the appendix, AdaWarp represents a special case of a more generalized adaptive filtering framework. Future work could investigate contextual differences beyond intensity differences through high-dimensional adaptive filtering. **Second**, projecting to high-dimensional space results in many zero-valued cells, with non-zero cells forming a submanifold [76]. Restricting grid manipulations, such as convolutional filtering, to this submanifold could further reduce computational overhead.

## 6 CONCLUSIONS

Our paper introduces AdaWarp, a novel neural network module for medical image registration that improves both accuracy-efficiency and accuracy-smoothness trade-offs by leveraging the piece-wise smooth prior. This prior is implemented through a learnable bilateral grid with guidance mapping, enabling accurate low-frequency approximations while preserving boundary details. Its success in two medical registration tasks highlights its broader applicability to similar problems. Moreover, AdaWarp transforms deformable registration into a keypoint detection task, with potential applications in segmentation tasks as briefly discussed in the appendix.

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

## A   PSEUDO-GT GENERATION

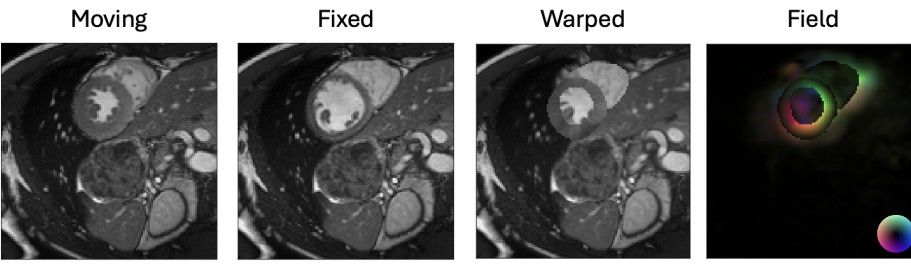

Figure 8: The first two figures depict the original moving and fixed images, while the third figure shows the warped image generated by DDIR. The final figure illustrates the deformation field, where colors correspond to the normalized direction and magnitude of displacement, as indicated in the reference circle at the bottom right.

The pseudo-ground truth (pseudo-GT) deformation field is generated using DDIR [31], a pioneering neural network designed to produce high-quality deformation fields that preserve discontinuities. DDIR employs region-specific masking to separate foreground and background areas, generating

separated deformation fields for each region through independent yet identical U-Net architectures [7]. The final deformation field is obtained by composing all the separated deformation fields, ensuring local smoothness while preserving edge discontinuities. Prior methods [77, 78] generate deformation fields directly from region-of-interest masks. However, these approaches overlook intensity variations across different places of a region, which can degrade registration accuracy.

The accuracy of segmentation masks significantly influences the performance of DDIR. In the original work, the masks were automatically generated by a neural network[79], which led to suboptimal results for producing high-quality deformation fields. However, when using ground truth masks, the Dice score between the warped moving image and the fixed image reaches 98%, allowing the corresponding deformation field to be considered a pseudo ground truth, see Fig. 8.

## B  CONNECTIONS WITH SELF-ATTENTION.

Here we show that in an arbitrary dimensional space, Lipschitz-constrained L2 self-attention [80] becomes a special case of this adaptive filtering approach, specifically when the key-query projection matrices satisfy $\mathbf{W}_q = \mathbf{W}_k$ [81].

$$\tilde{\mathbf{v}}_i = \sum_{j=1}^{n} e^{-\frac{1}{2}\|\mathbf{p}_i - \mathbf{p}_j\|^2} \mathbf{v}_j. \tag{5}$$

Eq. (5) represents high-dimensional Gaussian filtering that associates each position vector $\mathbf{p}_i$ with a corresponding value vector $\tilde{\mathbf{v}}_i$ to be filtered, then combines these values with others located at proximate positions. By substituting $\mathbf{p}_i$ with $\mathbf{x}_i^\top \mathbf{W}$ and $\mathbf{v}_j$ with $\mathbf{x}_j^\top \mathbf{W}_v$, the model is transformed into constrained self-attention as described in Eq. (6). In this formulation, self-attention is equivalent to high-dimensional Gaussian filtering. When projecting these Gaussian kernels back into the image space, each pixel is assigned a different weight, effectively implementing adaptive filtering. Replacing the Gaussian kernel with a learnable kernel $\mathcal{K}$ parametrized by $\theta$ extends Eq. (6) to adaptive high-dimensional filtering in Eq.(7), positioning constrained self-attention as a special case within this broader framework. Here, $\gamma$ is a normalization factor derived from the Softmax weights used in the attention mechanism.

$$\tilde{\mathbf{v}}_i = \frac{1}{\gamma_i} \sum_{j=1}^{n} e^{-\frac{1}{2}\|\mathbf{x}_i^\top \mathbf{W} - \mathbf{x}_j^\top \mathbf{W}\|^2} \mathbf{x}_j^\top \mathbf{W}_v. \quad (6) \quad \tilde{\mathbf{v}}_i = \frac{1}{\gamma_i} \sum_{j=1}^{n} \mathcal{K}_\theta(\|\mathbf{x}_i^\top \mathbf{W} - \mathbf{x}_j^\top \mathbf{W}\|^2) \mathbf{x}_j^\top \mathbf{W}_v. \quad (7)$$

Our AdaWarp is also a special case of Eq. (7). If we replace $\mathbf{x}_i^\top \mathbf{W} - \mathbf{x}_j^\top \mathbf{W}$ with $\mathbf{G}_i - \mathbf{G}_j$, where $\mathbf{G}_i = [\mathbf{G}_i^x, \mathbf{G}_i^y, \mathbf{G}_i^r]$, representing spatial coordinates $(\mathbf{G}_i^x, \mathbf{G}_i^y)$ along with the guidance map value $\mathbf{G}_i^r$ at that location, with $\mathcal{K}_\theta$ as the learnable convolutional filters, it is essentially the proposed adaptive filtering.

## C  TRAINING DETAILS

Hyperparameters are optimized via grid search, and all learning-based networks use the Adam optimizer with a 1e-4 learning rate and a polynomial decay scheduler (rate 0.9). For smoothness, we apply L2 regularization on deformation gradients ($\lambda = 0.01$ for cardiac, $\lambda = 1$ for abdomen). We use MSE for cardiac and local NCC for abdomen as dissimilarity losses, with an additional Dice loss for incorporating abdomen segmentation (training only). Cardiac models are trained for 500 epochs with a batch size of 4, while abdomen models are trained for 100 epochs with a batch size of 1.

### C.1  DESCRIPTION OF $\lambda$ VALUES

We performed a grid search with $\lambda = 0.01, 0.1, 1.0$, and 5.0. We found $\lambda = 0.01$ to be optimal for the ACDC dataset and $\lambda = 1.0$ for the abdomen dataset. Most baseline methods use the same parameters and training settings as AdaWarp. Below, we provide more details:

### C.1.1  ACDC DATASET

All learning-based methods adopt the same hyperparameters as AdaWarp, with $\lambda = 0.01$, MSE as the dissimilarity loss, and scaling-and-squaring with 7 steps for the diffeomorphic transformation

model. For Fig. 5, while keeping other hyperparameters the same, we vary computational complexity by adjusting the starting channel count in FourierNet, LKU-Net, and Ada-Cost, and by modifying the backbone of TransMorph (tiny, small, and normal).

## C.2 ABDOMEN DATASET

The abdomen dataset presents more challenges due to the large displacement problem. To clarify:

- FourierNet, VoxelMorph, TransMorph, and Ada-Cost use local NCC as the dissimilarity loss with $\lambda = 1.0$. ConvexAdam and SAMConvex also use $\lambda = 1.0$, employing MIND and segmentation feature maps, respectively, to compute the dissimilarity. All these methods adopt scaling-and-squaring with 7 steps for the diffeomorphic transformation model.

- For TextSCF, we follow its original implementation with $\lambda = 0.1$ and without the diffeomorphic transformation model. The $\lambda = 0.1$ version with integration is also presented in Fig. 7.

- Both LKUNet and LapIRN results in Table 2 use $\lambda = 1.0$ with the diffeomorphic transformation model.

## D  DISPLACEMENT FIELD MANIPULATION

This section mainly develops for application of keypoints-based lung CT registration. We briefly review deformable image registration (DIR), followed by derivation of displacement field manipulation via a bilateral grid. DIR aligns a moving image $\mathbf{I}_m$ with a fixed image $\mathbf{I}_f$ using spatial mapping $\phi(x) = x + \mathbf{u}(x)$, where $\mathbf{u}(x)$ is the displacement at $x$ in domain $\Omega \subset \mathbb{R}^{H \times W \times D}$. This warps $\mathbf{I}_m$ for voxel correspondence with $\mathbf{I}_f$, using linear interpolation for non-grid positions. Unsupervised learning estimates deformation field $\phi$ through a network $F_\theta$, optimizing weights $\theta$ via a composite loss function $\mathcal{L}$. This combines dissimilarity between $\mathbf{I}_m$ and $\mathbf{I}_f$, and deformation field smoothness:

$$\mathcal{L} = \mathcal{L}_{sim}(\mathbf{I}_f, \mathbf{I}_m \circ \phi) + \mathcal{L}_{sim}(\mathbf{G}_f^r, \mathbf{G}_m^r \circ \phi) + \lambda \mathcal{L}_{reg}(\phi), \tag{8}$$

where $\mathbf{G}_f^r$ and $\mathbf{G}_m^r$ are derived from $\mathbf{I}_f$ and $\mathbf{I}_m$ via guidance mapping. For cardiac registration, we use the loss function defined in Eq. (8). In lung registration, this is augmented with an additional Dice loss using segmentation masks and target registration loss based on keypoints.

### D.1  DISPLACEMENT FIELD MANIPULATION.

Using a bilateral grid, we can create a complete displacement field solely from keypoints. Given moving keypoint $\mathbf{p}_m \in \mathbb{R}^{N \times 3}$ and corresponding fixed keypoints $\mathbf{p}_f \in \mathbb{R}^{N \times 3}$, a sparse displacement field is formed by setting $\mathbf{u}[\mathbf{p}_f(i)] = \mathbf{p}_m(i) - \mathbf{p}_f(i)$ for each $i \in \{1, 2, ..., N\}$, where $\mathbf{p}_*(i)$ denotes the $i_{th}$ point in the set. For all other points $\mathbf{u}(x)$ is a zero vector. Given the guidance map $\mathbf{G}^r$ derived from either a guidance map generator or the raw image, the sparse displacement field $\mathbf{u}$ can be projected onto a bilateral grid as $\mathbf{\Gamma} = \mathcal{D}(\mathbf{u}, \mathbf{G}; \mathcal{K})$. Here, $\mathbf{G}$ includes mesh grids and $\mathbf{G}^r$, and $\mathcal{K}$ is a linear sampling kernel. The task then is to minimize the equation $\arg\min_{\tilde{\mathbf{\Gamma}}} \sum ||\mathrm{grad}(\tilde{\mathbf{\Gamma}})||^2$, subject to the constraint $\tilde{\mathbf{\Gamma}}(x) = \mathbf{\Gamma}(x)$, for every $x$ that $\mathbf{\Gamma}(x) \neq \mathbf{0}$. This is used to fill up the zero values in the grid. The optimization outlined in the equation can be efficiently executed through convolution and seamlessly integrated into neural network training.

### D.2  ZERO-SHOT CAPABILITY

In this section, we demonstrate that AdaWarp retains the functionality of traditional bilateral grids while being implemented in a more user-friendly PyTorch framework. We evaluate AdaWarp's zero-shot inference in propagating sparse displacement vectors to fill the entire image based on intensity, using the L2R-NLST dataset [82, 83], which contains paired low-dose helical lung CT images at inhale and exhale phases, along with sparse auto-generated keypoints. Details on propagation of sparse displacement vectors can be found in §D of the appendix, with more background details on bilateral grids in [36].

Among the evaluated methods, FourierNet+ and LKU-Net are specifically trained and fine-tuned for this task, whereas uniGradICON has been trained on various medical image datasets spanning

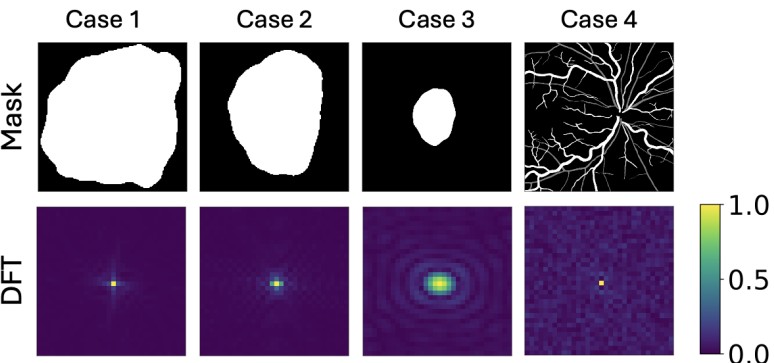

Figure 9: The first row presents lesion masks of varying sizes and a vessel mask. The second row displays the corresponding DFT magnitude spectra, normalized to the range (0,1) and resized to $32 \times 32$ to improve visualization clarity.

different anatomical regions and contrasts. Although uniGradICON's training data includes lung CT images, these were sourced from different studies. Unlike the benchmarks, our Ada-KPs model has not been trained on any lung CT dataset.

For quantitative evaluation, the Learn2Reg leaderboard [84] measures Target Registration Error (TRE) using anatomical landmarks (LM) manually labeled by experts and auto-generated keypoints (KPs). All results are obtained from the Learn2Reg leaderboard for the NLST task. As shown in Table 3, networks specifically trained on this dataset exhibit slightly better/lower TRE (LM) compared to untrained ones. However, AdaWarp achieves the lowest TRE (KP) due to the bilateral grid propagating from these KPs. The higher TRE (LM) may be due to the sparse nature of auto-generated KPs and the limited accuracy of keypoint correspondence between moving and fixed images. A denser, more accurate set of keypoints would likely reduce TRE (LM). As a result, AdaWarp allows for improved lung CT image registration by solely enhancing the precision of auto-generated keypoints, effectively transforming the deformable registration problem into a keypoint detection task.

| Model | TRE (LM) ↓ | TRE (KP) ↓ |
|---|---|---|
| FourierNet+ [13] | **1.55±0.24** | 0.82±0.13 |
| LKU-Net [11] | 1.55±0.26 | 0.75±0.13 |
| uniGradICON [52] | 2.07±0.43 | 1.26±0.28 |
| uniGradICON (IO) [52] | 1.77±0.29 | 0.98±0.18 |
| **Ada-KPs (Ours)** | 1.79±0.54 | **0.08±0.18** |

| Model | Dice (%)↑ | HD95↓ |
|---|---|---|
| Swin Encoder Upsample [50] | 84.73 | 17.98 |
| Swin-UNet [85] | 88.35 | 17.95 |
| PVT-CASCADE [86] | 88.51 | 16.55 |
| Trans-UNet [87] | 88.93 | 16.60 |
| **Ada-Swin (Ours)** | **90.02** | **15.16** |

Table 3: Quantitative evaluation of different models on the L2R-NLST dataset, highlighting top scores in bold. Metrics include TRE (LM) (mm) and TRE (KP) (mm), with a ↓ indicating that lower values are better.

Table 4: Quantitative evaluation of different models on the test set of ISIC2018 dataset, highlighting top scores in bold. Metrics include Average Dice (%) and HD95 (in pixels), with a down arrow indicating that lower values are better.

# E EXTENDING ADAWARP TO IMAGE SEGMENTATION

Similar to approximating deformation fields in the low-frequency range of the Fourier domain, segmentation masks exhibit piece-wise constant structures. As shown in Fig. 9, larger masks concentrate in the low-frequency range, while smaller masks spread their frequency components. For vessel masks, frequency components, except for the 0 component, are distributed more uniformly. This suggests that bilinear upsampling may suffice for segmenting large masks, but smaller masks require more careful handling.

We use skin lesion segmentation to validate our hypothesis, leveraging the ISIC2018 challenge dataset [88, 89] with 2,594 training images, 100 for validation, and 1,000 for testing. The model's ability to segment *piece-wise constant* masks aligns with its core design. During training, raw images are input with corresponding masks as output, without architectural modifications. We evaluate segmentation accuracy using Dice and HD95 (in pixels). For this task, we use Swin Transformer [50] as the encoder, with one model performing direct upsampling ("Swin Encoder Upsample")

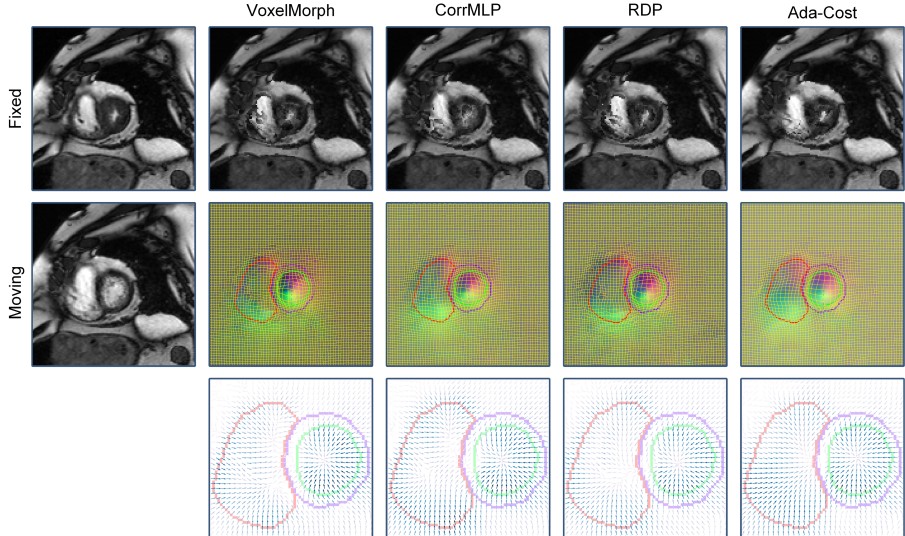

Figure 10: Qualitative results on the ACDC dataset. The first row shows the original fixed image and the warped moving images produced by each method. The second row displays the original moving image and the deformation fields in grid format for each method. The third row presents zoomed-in 2D vector fields projected onto the axial plane. Color contours indicate objects of interest, with different organs represented in distinct colors.

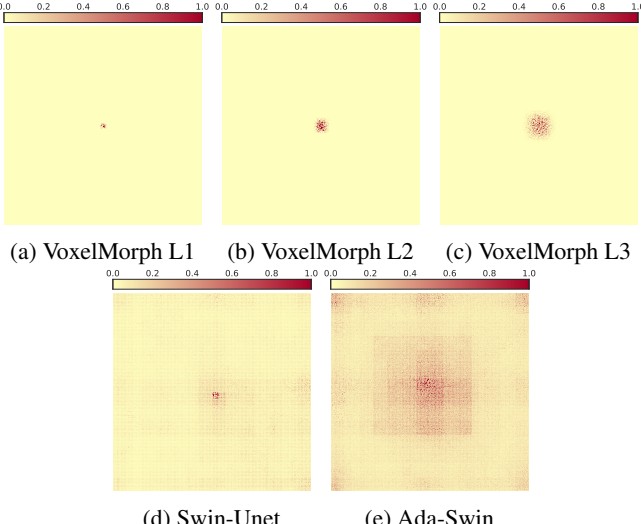

Figure 11: *Effective Receptive Field (ERF)* visualizations [26] across architectures: VoxelMorph (a, b, c), Swin-Unet [50, 85] (d), Ada-Swin (e). Darker and more widely spread regions indicate larger ERFs. Swin-Unet and Swin-Slicer feature maps are presented pre-softmax, while Unet utilizes encoder feature maps, with L3 to L1 showing increased spatial sizes via upsampling.

and another using AdaWarp for upsampling ("Ada-Swin"). Other baselines include Trans-UNet [87], Swin-UNet [85], and PVT-CASCADE [86]. Ada-Swin achieves the highest Dice (%) and lowest HD95 compared to state-of-the-art methods, demonstrating AdaWarp's versatility for medical imaging tasks. Additionally, Swin Encoder Upsample, while achieving the lowest Dice (%) at 84.73%, still maintains a reasonable score, supporting our spectrum analysis assumptions.

# F  QUALITATIVE RESULTS AND EFFECTIVE RECEPTIVE FIELD

## F.1  EFFECTIVE RECEPTIVE FIELDS (ERFS)

The low-resolution feature maps from deeper layers of neural networks inherently possess larger effective receptive fields (ERFs) than shallower layers. Please refer to the ERF visualization available via Fig. 11. Darker and more widely spread regions indicate larger ERFs. The details of ERF computation can be found in the seminal work[26]. Our key observations are as follows:

### F.1.1  LEVERAGING LARGE RECEPTIVE FIELDS

Subfigures (a), (b), and (c) illustrate feature maps from different encoder levels (L1: full resolution, L2: 1/2 downsampled, L3: 1/4 downsampled) from VoxelMorph. Deeper layers (e.g., L3) have larger ERFs, confirming that low-resolution features from deeper layers capture broader context. AdaWarp leverages the deepest encoder layer for the largest possible ERF. As shown in Table 4 (first row vs. last row), maintaining object boundaries (via AdaWarp) is essential for accuracy, as large ERFs alone are insufficient.

### F.1.2  EFFECTIVENESS OF ADAWARP OVER SWIN-UNET

We compared ERF heatmaps of Swin-Unet and Ada-Swin (pre-softmax feature maps of models used in Table 4). Both share identical encoders, differing only in the decoder (Swin-Unet uses a U-Net structure, while Ada-Swin uses AdaWarp). Ada-Swin shows larger ERF regions and achieves 1.89% higher accuracy than Swin-Unet.

