# OpenReview forum: "Efficient Adaptive Filtering for Deformable Image registration"
_ICLR.cc/2025/Conference — Submitted to ICLR 2025_

### Official Review · Reviewer_RMNZ · 2024-10-23

**Soundness:** 3
**Presentation:** 2
**Contribution:** 3
**Rating:** 6
**Confidence:** 4

**Summary:**

The paper presents AdaWarp, a novel architecture in medical image registration. The model introduces a piece-wise smooth (P-S) assumption, which exploits the smoothness of intensity variations within anatomical regions while preserving sharp boundaries between organs. This assumption is incorporated into the network through a differentiable bilateral grid, which allows for efficient edge-preserving filtering and reduces computational complexity.

**Strengths:**

1. The integration of the differentiable bilateral grid into the deep learning framework for image registration is highly innovative. It effectively addresses the limitations of traditional smoothness constraints, enabling the model to better handle complex and localized deformations.

2. The paper is well-structured, offering a clear explanation of the proposed methods. It provides detailed descriptions of the differentiable bilateral grid, encoder architecture, and adaptive filtering process. Visual aids, such as Figures 4 and 5, are particularly useful in clarifying complex comparisons.

3. This method presents a promising alternative for resolving the conflict between global smoothness and local deformations, potentially offering improved solutions in certain applications.

**Weaknesses:**

The weaknesses of the paper are primarily in the literature review and experimental sections, which lack sufficient references and baseline comparisons, as well as visual results. These limitations are why I rated the paper as "fair" in terms of Presentation and Soundness.


1. The paper needs more references in the literature review. The current review only discusses works that do not address the conflict between global smoothness and local deformations. However, this is not the first paper to tackle this problem. Research such as multi-scale registration and patch-wise registration also offers relevant solutions. While these methods may not explicitly incorporate the piece-wise smooth prior, they still manage local deformations while maintaining overall smoothness. The authors should include these references in the background and select baselines from this body of work to show that the proposed method offers a superior solution to the problem.

2. The experiments do not adequately support the claimed advantages of the proposed method. While the paper argues that the model can generate sharp boundaries between organs by incorporating the P-S assumption, it fails to provide visual results to substantiate this key contribution. Relying solely on numerical metrics like Dice, HD95, and SDlogJ does not clearly demonstrate that the model’s output preserves sharp boundaries.

3. The writing in the experiments section is somewhat disorganized. The authors employ significantly different model structures and training strategies, including both unsupervised and semi-supervised approaches (which require further clarification), depending on the dataset. This inconsistency raises concerns about the generalizability of the model across different tasks. Additionally, the experiments lack ablation studies, which are necessary to demonstrate the effectiveness of each component in the proposed methods.

**Questions:**

1. Why were different model structures used for different datasets? What would be the result of using Ada-Res on the Abdomen CT dataset and Ada-Cost on the ACDC dataset? A comparison of these model structures across datasets could help demonstrate their generalizability and clarify why different architectures were chosen for each.

2. In the Abdomen CT dataset, Ada-Cost uses “the same segmentation model for feature extraction.” Was this segmentation model pre-trained? If so, this would make Ada-Cost a semi-supervised registration model. Comparing it with other unsupervised deep learning-based methods would be unfair. Additionally, how exactly was the segmentation model integrated into your model’s structure? Does it replace the "guidance map generator," or is it incorporated elsewhere in the architecture?

3. More references, more baselines and visual evaluations of warped images and warped segmentation masks would be highly valuable. Providing such visual results would help demonstrate the effectiveness of your method in producing sharp boundaries, which cannot be fully illustrated through numerical metrics alone.

4. I would greatly appreciate it if the paper could provide information on the inference and training time of the proposed method. This data would offer more valuable insights into the computational efficiency of the model.

5. Another concern is that the authors selected "interpretability and explainable AI" as Primary Area. I’m not sure if this is appropriate since there is no work about interpretability of proposed method.

---

> ### Author Response · Authors · 2024-11-25
> **Part 1: Addressing [W1] and [Q1]**
>
> Dear Reviewer RMNZ,
>
> We sincerely appreciate your comments and suggestions. Below, we address your concerns listed in the weakness section as [W1], [W2], and [W3], along with your questions. Note that some questions overlap with the weaknesses, so we have combined them in our responses.
>
> [W1] Reply: Thank you for raising this point. We have added more references to the literature review in the revised manuscript and included two recent multi-scale approaches: CorrMLP [1], also mentioned by Reviewer MgEu, and RDP [2]. Both methods demonstrate improved performance compared to existing baselines in Tables 1 and 2. Additionally, we note that several existing baselines, such as MemWarp, LapIRN, and discrete optimization-based methods, also employ image pyramids for multi-scale processing.
> Below are the results of the added methods compared to ours for reference:
>
> **Cardiac Dataset:**
> | Model          | Avg. (%) | RV (%) | LVM (%) | LVBP (%) | HD95 (mm) ↓ | SDlogJ ↓  |
> |----------------|----------|--------|---------|----------|-------------|-----------|
> | Initial        | 58.14    | 64.50  | 48.33   | 61.60    | 11.95       | -         |
> | CorrMLP        | 77.58    | 74.84  | 75.68   | 82.21    | 9.23        | 0.052     |
> | RDP            | 77.62    | 74.70  | 75.95   | 82.20    | 9.15        | 0.053     |
> | Ada-Res (Ours) | 79.20    | 78.14  | 76.31   | 83.15    | 8.33        | 0.050     |
>
>
> **Abdomen Dataset:**
> | Model             | Type  | Dice (%) | HD95 (mm) ↓ | SDlogJ ↓  |
> |-------------------|-------|----------|-------------|-----------|
> | Initial           | -     | 30.86    | 29.77       | -         |
> | CorrMLP           | L     | 56.58    | 20.40       | 0.16      |
> | RDP               | L     | 58.77    | 20.07       | 0.22      |
> | Ada-Cost (Ours)   | L&D   | 62.74    | 15.03       | 0.12      |
>
>
> [Q1] Reply: Thank you for raising this point. Initially, we developed Ada-Res for the ACDC dataset to achieve strong performance. However, applying Ada-Res to the abdomen dataset revealed suboptimal performance due to inherent challenges like the aperture and large displacement problems.
> To address these, we researched solutions tailored to abdomen datasets and found that incorporating image pyramids and discrete optimization was more effective. Leveraging the flexibility of AdaWarp, we integrated these approaches into Ada-Cost.
> To verify generalizability, we tested Ada-Cost on the ACDC dataset and observed improved Dice scores but slightly degraded HD95 compared to Ada-Res. We have updated the manuscript to reflect Ada-Cost results on ACDC and ensure consistency in model structure across datasets.
> Below are the results of the Ada-Cost applied to ACDC dataset in comparison to initial Ada-Res:
>
> **Cardiac Dataset:**
> | Model            | Avg. (%) | RV (%) | LVM (%) | LVBP (%) | HD95 (mm) ↓ | SDlogJ ↓  |
> |------------------|----------|--------|---------|----------|-------------|-----------|
> | Initial          | 58.14    | 64.50  | 48.33   | 61.60    | 11.95       | -         |
> | Ada-Res (Ours)   | 79.20    | 78.14  | 76.31   | 83.15    | 8.33        | 0.050     |
> | Ada-Cost (Ours)  | 79.82    | 77.58  | 77.95   | 83.92    | 8.98        | 0.050     |
>
> In addition, we would like to clarify that while many studies use the same backbone network across datasets to demonstrate generalizability, **AdaWarp is a flexible network module** rather than a full backbone that can be easily integrated into existing frameworks.
>
> - For instance, we incorporated AdaWarp into ConvexAdam as a bilateral filter (Table 2). Although the improvement is limited due to the non-learnable nature of ConvexAdam, this demonstrates AdaWarp's flexibility.
>
> - As discussed in the manuscript, AdaWarp can also be integrated into other registration frameworks, such as **zero-shot lung CT registration** based on keypoints and image intensities, with preliminary results shown in Table 3. Furthermore, AdaWarp can be applied to **medical image segmentation** tasks, with preliminary results provided in Table 4.
>
> [1] Meng, M., Feng, D., Bi, L. and Kim, J., 2024. Correlation-aware Coarse-to-fine MLPs for Deformable Medical Image Registration. CVPR 2024.
>
> [2] Wang, H., Ni, D. and Wang, Y., 2024. Recursive deformable pyramid network for unsupervised medical image registration. TMI 2024.

---

> ### Author Response · Authors · 2024-11-25
> **Part 2: Addressing [W3] and [Q2]**
>
> [W3] Reply: As stated in our response to Q1, AdaWarp is a plug-and-play module rather than a full backbone network, allowing flexibility in selecting the most suitable backbone based on dataset specifics to achieve higher accuracy. Contrary to the concern that using different network structures for different datasets might lead to inconsistencies or generalizability issues, we believe this demonstrates AdaWarp’s adaptability across various architectures and datasets with differing modalities and input constraints.
>
> **Network Architectures:**
> - Both Ada-Res and Ada-Cost perform well on the ACDC dataset, as it is relatively simpler compared to the Abdomen dataset. While ACDC has local discontinuities and sliding motions, displacements are smaller (up to 15 voxels) than in the Abdomen dataset (up to 60 voxels).
> - On the Abdomen dataset, Ada-Res achieves suboptimal performance due to the larger displacements. However, AdaWarp can be easily adapted and integrated into a more suitable backbone to optimize task-specific performance such as a combination of image pyramid and discrete optimization cocnept, highlighting its generalizability rather than inconsistency.
>
> **Input Constraints and Modalities:**
> - The ACDC dataset is used for unsupervised learning, where segmentation masks are not used during training or testing.
> - The Abdomen dataset is used for semi-supervised learning, where segmentation masks are provided only during training as auxiliary loss supervision and are not used during testing.
>
> The unsupervised and semi-supervised settings remain consistent across all methods. Rather than reflecting inconsistency, this setup demonstrates AdaWarp’s generalizability across modalities (MRI in ACDC, CT in Abdomen) and input constraints (unsupervised in ACDC, semi-supervised in Abdomen).
>
> **For the ablation studies**, the key variables in AdaWarp are the spatial sampling rate ($s_s$) and the range sampling rate ($s_r$), which we have discussed in Section 5.1 of the discussion. Additionally, we analyzed the impact of varying $\lambda$ in Figure 6 and thoroughly examined the accuracy-efficiency and accuracy-smoothness trade-offs. Please let us know if there are specific components or parameters you would like us to explore further.
>
> [Q2] Reply: Thank you for pointing this out. We acknowledge that the description was unclear, and we clarify it here. The segmentation network used is a pretrained network with weights from [1], and the pre-softmax feature maps from its output are used as feature maps for SAMConvex [2]. These feature maps are specifically used to compute the dissimilarity loss in SAMConvex. For details, please refer to their respective papers. Briefly, ConvexAdam [3] and SAMConvex are instance-optimization-based methods, requiring iterative optimization for each input pair rather than amortized optimization. ConvexAdam uses MIND [4] as feature maps, while SAMConvex relies on a contrastively pretrained network (which we lack), so we substitute the segmentation feature maps.
>
> - **How we use the segmentation feature maps:** We use the feature maps of moving and fixed images as input to Ada-Cost but continue to use raw images for dissimilarity loss computation. Thus, the feature map does not replace the guidance generator. For more details, refer to our reply to [C] for Reviewer UMDT.
> - **Semi-supervised vs. Unsupervised:** Yes, this makes Ada-Cost a semi-supervised method on the abdomen dataset. However, other methods on this dataset also use auxiliary segmentation loss for supervision, so we believe this is fair. For clarity, the Ada-Cost used on the ACDC dataset relies solely on raw images as input, without any segmentation network or segmentation loss.
>
> [1] Liu, J., Zhang, Y., Chen, J.N., Xiao, J., Lu, Y., A Landman, B., Yuan, Y., Yuille, A., Tang, Y. and Zhou, Z., 2023. Clip-driven universal model for organ segmentation and tumor detection. ICCV 2023.
>
> [2] Li, Z., Tian, L., Mok, T.C., Bai, X., Wang, P., Ge, J., Zhou, J., Lu, L., Ye, X., Yan, K. and Jin, D., 2023, October. Samconvex: Fast discrete optimization for ct registration using self-supervised anatomical embedding and correlation pyramid. MICCAI 2023.
>
> [3] Siebert, H., Großbröhmer, C., Hansen, L. and Heinrich, M.P., 2024. ConvexAdam: Self-Configuring Dual-Optimisation-Based 3D Multitask Medical Image Registration. TMI 2024.
>
> [4] Heinrich, M.P., Jenkinson, M., Papież, B.W., Brady, S.M. and Schnabel, J.A., 2013. Towards realtime multimodal fusion for image-guided interventions using self-similarities. MICCAI 2013.

---

> > ### Comment · Reviewer_RMNZ · 2024-11-25
> > **Reply to author's comment**
> >
> > Thank you for addressing many of the concerns raised in the initial review. I appreciate the effort the authors have put into improving the manuscript. While several issues have been resolved, I still find the paper's motivation (or "story") lacking experimental validation. As mentioned in Weakness 2 and Question 3, there remains insufficient evidence to demonstrate that the proposed method can achieve registration results with sharp boundaries(no visual results). This lack of solid experimental support undermines the alignment between the proposed method and the claimed contributions of the paper.
> >
> > To clarify, I am not suggesting that the method itself lacks merit—it is indeed novel within the registration field and holds potential for future impact. However, the current experimental section of the paper does not convincingly support its claims, which limits its immediate contribution.
> >
> > After careful consideration, I have decided to maintain my score.

---

> > > ### Author Response · Authors · 2024-11-26
> > > **Reply to Reply to author's comment**
> > >
> > > Dear Reviewer RMNZ,
> > >
> > > Thank you for your feedbacks. We would like to provide a gentle reminder of the ICLR submission-review timeline, as outlined in the email from the ICLR program chairs.
> > >
> > > **Timeline:**
> > > We are still in the discussion phase and have not yet responded to all your comments or uploaded our revised manuscript. The timeline is as follows:
> > > - **November 27th**: Last day to upload a revised PDF. After this date, only forum replies are allowed (no manuscript changes).
> > > - **December 2nd**: Last day for reviewers to post messages to authors (six-day extension).
> > > - **December 3rd**: Last day for authors to post messages on the forum (six-day extension).
> > >
> > > > *"As mentioned in Weakness 2 and Question 3, there remains insufficient evidence to demonstrate that the proposed method can achieve registration results with sharp boundaries (no visual results)."*
> > >
> > > We appreciate your patience as we work to address everyone’s every concern, including your specific feedback on qualitative results. We are actively working on this and will provide updates as soon as possible. Thank you for your understanding.

---

> ### Author Response · Authors · 2024-11-29
> **Part 3: Addressing [W2] and [Q3-5]**
>
> [W2&Q3] Reply:
> We have added more references to image registration baselines in the related work and included comparisons with two recent multi-scale approaches, CorrMLP and RDP, both quantitatively and qualitatively. For qualitative results, please refer to Figure 4 and Figure 10 in the revised manuscript. A brief summary of key observations is provided in the third point of the **Author Response Summary**.
>
> [Q4] Reply:
> We apologize for not recording detailed timing information in the original manuscript. While timing depends on the model's parallel or sequential structure, inference and training times are generally correlated with the Multi-Adds (G) used. Since all models were trained under the same configuration, Multi-Adds (G) serves as a proxy for timing comparison. Additionally, we note that Ada-Cost converges faster than other methods. For example, on the abdomen dataset, Ada-Cost typically converges within 20–30 epochs, whereas other methods may require 50–80 epochs.
>
> [Q5] Reply:
> Thank you for raising this question, which ties closely to the qualitative results. The interpretability of AdaWarp can be summarized as follows:
> - **Physical Prior**: AdaWarp is motivated by the piece-wise smooth assumption, prevalent in medical imaging. Unlike other learning-based methods that treat the model as a black box, AdaWarp introduces interpretability by constructing cost volumes after feature extraction and applying edge-preserving filtering to reflect local discontinuities. The qualitative results confirm that these discontinuities are effectively preserved.
> - **Simpler Architecture**: Unlike methods relying on complex modules like self-attention or recurrent networks, AdaWarp uses only a handful of convolutional layers. These simpler mechanisms are more interpretable and easier to understand compared to advanced neural network components.

---

> > ### Comment · Reviewer_RMNZ · 2024-11-29
> > **Reply to Part 3: Addressing [W2] and [Q3-5]**
> >
> > Thank you for your effort and for providing additional figures. I truly appreciate the improvements made to the manuscript. However, the visual results do not clearly demonstrate that the proposed method achieves significantly sharper boundaries compared to the baselines. For instance: Figure 4: While there is a slight improvement in boundary sharpness at the top of the liver (visible only after zooming), the boundary constraints appear to fail in regions like the ribs when compared to ConvexAdam. Figure 10: I struggled to observe any noticeable differences in boundary sharpness between the proposed method and the baselines.
> >
> > Suggestions: It is unclear whether these subtle differences stem from the choice of images for visualization or limitations in the method itself. If it is the former, I recommend selecting a subject that better highlights the advantages of your approach. If it is the latter, consider revising your narrative to align more closely with the actual performance of the method. Either option would be acceptable.
> >
> > Additionally, I suggest visualizing the warped images alongside correlated metrics with zoomed-in views to make the results more evident. This approach would make it easier for readers to understand your point. You might find Figure 5 in "NODEO: A Neural Ordinary Differential Equation Based Optimization Framework for Deformable Image Registration" and Figure 3 in "A Plug-and-Play Image Registration Network" to be helpful references for presenting such visualizations.
> >
> > Overall, the paper is progressing well. If you can address these points, I would be happy to reconsider and potentially raise my score.

---

> > > ### Author Response · Authors · 2024-11-30
> > > **Reply to Reply to Part 3: Addressing [W2] and [Q3-5]**
> > >
> > > Dear Reviewer RMNZ,
> > >
> > > Thank you for your valuable feedback. We would like to clarify and ensure that we have correctly understood your comments. Please feel free to correct us if there are any misunderstandings.
> > >
> > > **Main Claims**:
> > > While we understand that you are particularly focused on the expectation of **sharp boundaries** in the proposed method's results, we want to emphasize that **we did not claim in either the original or revised manuscript that the proposed method guarantees sharp boundaries in the warped moving image or the displacement field**.
> > >
> > > Our main focus remains on demonstrating that the proposed method achieves a piece-wise smooth displacement field and has the ability to preserve local discontinuities. Although we observed sharp boundaries in the produced displacement field (as described in Section 4.2.2 of the revised manuscript), this is not a specific claim of the manuscript or in the OpenReview discussion.
> > >
> > > **Piece-wise Smooth**:
> > > - To help you and potential readers better understand, the piece-wise smooth assumption does not inherently imply sharp boundaries. While sharp boundaries in a displacement field indicate smooth regions divided by them, a piece-wise smooth field does not necessarily exhibit sharp boundaries.
> > > - For example, consider two adjacent organs connected by tissue. While their intensities within each organ may differ, the movement across their boundary can remain smooth because of the connective tissue. In this case, the piece-wise smooth assumption does not result in sharp boundaries.
> > > - Another example of sharp boundaries in a displacement field occurs when sliding motion happens between an organ and the surrounding abdominal cavity, such as between the liver and kidney, as demonstrated in the qualitative results.
> > > - If the displacement field remains smooth within each respective region while allowing discontinuities across boundaries, it can effectively **reduce artifacts**. This is demonstrated by the fact that other baseline methods produce multiple shrinking centers in the displacement field of the right ventricle, whereas Ada-Cost produces only one center.
> > >
> > > To better address your concerns, we would like to seek clarification on the following points:
> > >
> > > **Sharp Boundary Results**:
> > >    - You have mentioned "sharp boundaries" multiple times in our discussion, but the term seems ambiguous. Could you clarify what you specifically mean by sharp boundaries? Are you referring to sharp boundaries in the warped moving image or in the displacement field?
> > >    - While we acknowledge that the integrity of ribs warped by Ada-Cost may not be better than ConvexAdam, we attribute this to the displacement field around the rib and background being excessively sharp, rather than insufficiently sharp, leading to rib shrinking. Thus, our question is: how do you define or measure the sharpness of these boundaries? Specifically, what criteria determine whether one result is *sharper* than another?
> > >
> > > We hope these questions help clarify your perspective so we can address your concerns more effectively. Once we receive your feedback, we are happy to provide additional qualitative results if necessary. Thank you again for your valuable insights.

---

> > > > ### Comment · Reviewer_RMNZ · 2024-11-30
> > > >
> > > > In your manuscript, the concept of sharp boundaries stems from statements such as:
> > > > "Previous studies have shown that incorporating prior knowledge improves the.......distinct boundaries often exist between organs and the background or neighboring organs........while clear and well-defined boundaries are formed by intensity differences between these regions (see Fig. 1, columns 1&2). These consistent intra-region smoothness and inter-region boundaries indicate that certain medical images exhibit piece-wise smooth structures.”
> > > > If I understand correctly, your P-S assumption is based on the idea that within organs, the registration field should be smooth, while at the boundaries of organs, there should be sharp transitions. In my review, when I refer to "sharp boundaries," I am aligning with the terminology you use—"clear, well-defined boundaries."
> > > >
> > > > My main confusions:
> > > > 1. Your P-S assumption is motivated by the prior knowledge that "distinct boundaries often exist between organs and the background or neighboring organs." However, your results do not guarantee the preservation of these boundaries. If the P-S assumption inherently does not guarantee sharp boundaries, why are boundaries a key component of your prior? Furthermore, you mention related works “To address these discontinuities, some works have employed bilateral filters [27, 28], which preserve edges and improve registration performance in the presence of local discontinuities”. It seems inconsistent that these prior approaches preserve boundaries, but your method, which should be better, cannot guarantee them.
> > > > 2. Based on your recent reply, I am also unclear about how you define "local discontinuities" in your work. My understanding is that local discontinuities refer to abrupt changes in the displacement field, often representing anatomical or structural boundaries. If your method is designed to preserve local discontinuities, how can it fail to guarantee sharp boundaries?
> > > >
> > > > Regarding the evaluation of sharp boundaries, I acknowledge that it can be challenging to quantify this numerically. However, at the very least, the manuscript should provide qualitative evidence where the proposed method demonstrates a clear advantage over the baselines. For example, you could present cases where other baselines fail to preserve sharp boundaries (e.g., instances where organs appear fused together), and your method successfully delineates clear boundaries between those organs.
> > > >
> > > > I hope this reply effectively conveys my concerns. If I have misunderstood the logic or interpretation of your paper, please do not hesitate to point it out. I am open to clarifications and look forward to better understanding your work.

---

> > > > > ### Author Response · Authors · 2024-11-30
> > > > >
> > > > > Dear Reviewer RMNZ,
> > > > >
> > > > > Thank you for your prompt response. We now have a clearer understanding of the source of confusion and believe the following clarification will help resolve it. Before getting into specific responses, we would like to establish some key points:
> > > > >
> > > > > 1. **Definition of Terms**:
> > > > >
> > > > > We believe that the terms "local discontinuities" or "sharp boundaries" refer to features in the **displacement field** rather than in the warped images. If you have a different interpretation, please let us know.
> > > > >
> > > > > 2. **Illustration of Displacement Field Discontinuities**:
> > > > >
> > > > > Please refer to [this figure](https://ibb.co/7J8hQrf), which demonstrates the desired behaviors of a discontinuous displacement field:
> > > > >    - **(a) Local Homogeneity**: Smooth displacement vectors within an organ, representing the expected uniform motion.
> > > > >    - **(b) Varying Magnitudes in Similar Directions**: Displacement vectors with different magnitudes but similar directions, illustrating soft tissue moving against rigid structures.
> > > > >    - **(c) Sliding Boundary Conditions**: Displacement vectors on opposite sides of a boundary moving in opposite directions, depicting sliding motions between adjacent organs or between organs and the background.
> > > > >
> > > > > 3. **Piece-wise Smoothness vs. Sharp Boundaries**:
> > > > >
> > > > > The ability to preserve discontinuities does not necessarily imply the presence of sharp boundaries in the produced deformation field. Most deformations in the human body are elastic, meaning they are generally continuously differentiable and invertible. Even for adjacent organs with sharp boundaries in terms of image intensity, the displacement field may remain smooth and continuous. **In essence, clear and well-defined boundaries in image intensities do not inherently result in displacement discontinuities; only motions between these objects and the background can lead to displacement discontinuities.**

---

> > > > > > ### Author Response · Authors · 2024-11-30
> > > > > > **Official Comment by Authors (cont'd)**
> > > > > >
> > > > > > Please let us know if you do not agree with the above basic logic. Below, we address your detailed points:
> > > > > >
> > > > > > 1. **The Discontinuities Produced by Ada-Cost:**
> > > > > >    Figure 4 in the revised manuscript clearly demonstrates that Ada-Cost produces discontinuities that other methods fail to capture, covering both types of discontinuities outlined in the **Illustration of Displacement Field Discontinuities**.
> > > > > >
> > > > > >    To assist in visualization, we have marked these discontinuities in the figure via this [link](https://ibb.co/GP9G25s). Type 2 discontinuities occur less frequently than type 3. Specifically:
> > > > > >    - The region in the **red box** contains type 2 discontinuities.
> > > > > >    - The region in the **green box** contains type 3 discontinuities.
> > > > > >
> > > > > > In these marked regions, all other methods fail to capture such discontinuities, further highlighting Ada-Cost's ability to preserve local discontinuities **effectively, though not perfectly**.
> > > > > >
> > > > > > 2. **Our Observations/Prior Knowledge**:
> > > > > >    - The following statement reflects an observation based on medical imaging: “Distinct boundaries often exist between organs and the background or neighboring organs... while clear and well-defined boundaries are formed by intensity differences between these regions (see Fig. 1, columns 1&2). These consistent intra-region smoothness and inter-region boundaries indicate that certain medical images exhibit piece-wise smooth structures.”
> > > > > >    - **Please distinguish observation vs. technical innovation claim**.  This observation is based on what our clinicians commonly encounter in daily medical scans and does not imply that our method can guarantee sharp boundaries. Achieving guaranteed **sharp boundaries whenever necessary**, would indeed be extremely challenging and represent a big breakthrough, which we do not claim in this work.
> > > > > >
> > > > > > 3. ** P-S Assumption and Sharp Boundaries**:
> > > > > > “If the P-S assumption inherently does not guarantee sharp boundaries, why are boundaries a key component of your prior?” We believe this reflects a misunderstanding of basic logic rather than an issue with the proposed method. Allow us to illustrate this with an analogy:
> > > > > >    **If the ICLR conference inherently does not guarantee paper acceptance, why is your paper still one of the submissions to this venue?**
> > > > > >    In essence, by incorporating the prior, our model has a **better capability** of preserving sharp boundaries compared to other learning-based methods. However, this **does not imply that it can achieve 100% success** in capturing all sharp boundaries. The difficulty of the abdomen dataset, as indicated by Dice scores below 70% across all methods, demonstrates that certain correct displacements are inherently missed.
> > > > > >
> > > > > > 4. “To address these discontinuities, some works have employed bilateral filters [27, 28], which preserve edges and improve registration performance in the presence of local discontinuities.”
> > > > > >    We do not see any inconsistencies between these prior approaches and ours, though we agree the sentence could be clarified for better readability.
> > > > > >
> > > > > >    Revised sentence:
> > > > > >    “To address these discontinuities, some works have employed bilateral filters [27, 28], which **help** preserve edges and improve registration performance in the presence of local discontinuities.”
> > > > > >
> > > > > >    We believe the aim of our proposed method overlaps with these prior approaches, as we extend the neural network’s ability to **help** preserve edges or sharp boundaries, thereby improving registration performance. However, our method goes beyond traditional approaches, as our filters to the bilateral grid are end-to-end trainable and learnable. The benefits of using learnable adaptive filters are demonstrated in Table 2. For example, while ConvexAdam applies traditional bilateral filters and shows a slight increase in Dice, the improvement is marginal compared to the gains achieved by Ada-Cost.
> > > > > >
> > > > > > 5. “If your method is designed to preserve local discontinuities, how can it fail to guarantee sharp boundaries?”
> > > > > >    This reflects a logical misunderstanding rather than an issue with the proposed method. Allow us to provide an analogy:
> > > > > >    **If your paper is written to get accepted by ICLR, how can it fail to guarantee paper acceptance?**
> > > > > >
> > > > > >    Similarly, while our method is designed to preserve local discontinuities and improve sharp boundary preservation compared to other learning-based methods, it does not guarantee sharp boundaries in every case. This is due to the inherent complexity and challenges of the datasets, such as the abdomen dataset, where no method achieves perfect performance.
> > > > > >
> > > > > > In conclusion, we understand that some details of the paper may be challenging to fully grasp, and we are committed to addressing every single one of your concerns to ensure clarity and a better understanding.

---

> > > > > > > ### Comment · Reviewer_RMNZ · 2024-11-30
> > > > > > >
> > > > > > > I apologize if my previous comments misused the term "guarantee"—that was not my intention. If I understand correctly, while applying the P-S assumption may help achieve clearer boundaries in the warped image, relying on it alone is insufficient. Instead, your contribution focuses on preserving displacement field discontinuities, and your evaluation is centered at that level, rather than extending directly to boundary clarity in the image domain.

---

> ### Author Response · Authors · 2024-11-30
>
> Thank you for your understanding.
> We acknowledge that it was our oversight to pre-assume that readers would share our consensus that discussions about image registration discontinuities commonly refer to the displacement field. This should have been made clearer in the manuscript.
>
> In addition, we would like to emphasize that a better ability to preserve local discontinuities is only one of the benefits of incorporating the P-S prior.
> As the name "piece-wise smooth" suggests, it involves two key aspects:
> 1. The differences between pieces, reflected by the boundaries.
> 2. The smoothness within each piece.
>
> As demonstrated in Figure 10 of the revised manuscript on the cardiac dataset, Ada-Cost produces more realistic displacement fields. While all methods generate smooth fields within the left ventricle blood pool and left ventricle myocardium, only Ada-Cost produces a smooth field in the right ventricle by having a single shrinking center, matching realistic cardiac motion. This is further validated by the increase in the Dice score for the right ventricle achieved by both Ada-Cost and DeBG, which also incorporates the P-S prior.

---

> > ### Comment · Reviewer_RMNZ · 2024-11-30
> >
> > Thanks to the authors for their detailed and insightful response to the concerns I raised. After thoroughly considering your explanations, I have decided to increase my original score.

---

> ### Author Response · Authors · 2024-12-01
>
> Dear Reviewer RMNZ,
>
> Thank you for your thoughtful consideration of our detailed responses. We greatly appreciate the time and effort you have dedicated to reviewing our work and revisiting your assessment. While we are grateful for the score increase, we would like to further emphasize how our work contributes to and benefits both the medical imaging and learning representation community, in the hope that you might consider **an even higher score**.
>
> 1. **Advancing the Field of Medical Image Registration**:
>    - **Addressing a Critical Need**: Our work fills a significant gap in medical image registration by seamlessly incorporating the piece-wise smooth (P-S) prior into an end-to-end trainable framework. This advancement not only enriches the theoretical foundations of registration algorithms but also provides a practical tool that the community can leverage for more accurate and reliable image alignment.
>    - **Methodological Innovation with Broad Applicability**: Beyond registration, our method builds up connetctions of learnable adaptive filtering with self-attention and gated attention mechanism. We believe this approach has the potential to serve as **a building block for next-generation neural networks**. Its adaptability and efficiency make it suitable for various image processing tasks, potentially inspiring new research directions and applications beyond the medical imaging community.
>
> 2. **Empowering the Community through Empirical Validation**:
>    - **Setting a New Performance Benchmark**: Our results demonstrate that Ada-Cost achieves state-of-the-art performance, both in quantitative metrics and in producing realistic displacement fields. By aligning with realistic cardiac motion, as shown in Figure 10, our method provides a reliable benchmark that others in the community can build upon.
>    - **Simplicity Enhancing Accessibility**: Despite using a very simple architecture consisting of only a handful of convolutional layers, Ada-Cost surpasses much more complex models. Its computational efficiency and ease of implementation make it accessible for researchers and practitioners, facilitating wider adoption and fostering further innovation.
>    - **Handling Complex Clinical Scenarios**: The ability of Ada-Cost to preserve local discontinuities, critical in datasets with sharp boundaries and sliding motions like the abdomen dataset, addresses significant challenges in medical imaging. This capability can improve the accuracy of diagnoses and interventions, directly benefiting clinical outcomes.
>
> 3. **Catalyzing Future Research and Clinical Applications**:
>    - **Foundation for Advanced Models**: By establishing a stronger baseline for learning-based registration models, Ada-Cost serves as a foundational building block for developing more powerful and advanced image registration frameworks.
>    - **Extending Impact Beyond Registration**: We have demonstrated preliminary results extending Ada-Cost to other challenging tasks, such as keypoint-based lung motion estimation and medical image segmentation. Additionally, our preliminary derivation on connections of adaptive filtering with self-attention showcases the method's potential to contribute to the development of **next-generation neural networks**. These extensions highlight the method's adaptability and its potential to address diverse challenges in medical imaging, ultimately benefiting patient care.
>
> 4. **Collaborative Engagement and Community Benefit**:
>    - **Open Dialogue and Transparency**: Throughout this discussion, we have engaged thoroughly and openly with your concerns, fostering a constructive dialogue that we believe strengthens the work and its presentation.
>    - **Enhancing Clarity for Community Adoption**: We have updated the manuscript to improve clarity and ensure that all key claims are well-supported by evidence, both qualitatively and quantitatively. By doing so, we aim to make our work more accessible and beneficial to the community, encouraging others to build upon our findings.
>
> We sincerely thank you for already increasing your score, which reflects your recognition of the novelty, simplicity, and practical impact of our work. Given its potential to serve as a foundation for further advancements and its broader applicability to critical tasks in medical imaging, we kindly ask you to consider these points for further raising your score, as we believe our contributions make a substantial impact on the community.
>
> Thank you once again for your thoughtful feedback and engagement with our work.

---

> > ### Comment · Reviewer_RMNZ · 2024-12-02
> >
> > I have no objection to this paper being accepted at ICLR 2025. However, in my assessment, the paper’s current state falls between a 6 and a 7, but does not quite reach an 8 based on my standards. While the work shows potential, I feel that both the contribution and the writing do not meet the level of quality required for an 8.
> >
> > Since the scoring system does not allow a 7, I can only justify maintaining my score at a 6. I cannot raise my score simply because I wish for the paper to be accepted. Therefore, after careful consideration, I have decided to maintain my score.

---

> > > ### Author Response · Authors · 2024-12-02
> > >
> > > Dear Reviewer RMNZ,
> > >
> > > Thank you for taking the time to provide thoughtful feedback and for supporting the potential acceptance of our paper at ICLR 2025. We understand and respect your reasoning for maintaining your score. Your comments have been very helpful, and we’ll keep working to improve both the contributions and the writing in future work.
> > >
> > > Best,
> > > The Authors

---

### Official Review · Reviewer_UMDT · 2024-10-30

**Soundness:** 2
**Presentation:** 2
**Contribution:** 3
**Rating:** 3
**Confidence:** 4

**Summary:**

The paper proposes a novel method to utilise prior knowledge (piece-wise smooth assumption) to enhance learning based registration striking a balance between computational complexity and accuracy. The performance is evaluated on a cardiac and an abdominal dataset.

**Strengths:**

The paper presents AdaWarp, a novel method that integrates the piece-wise smoothness assumption enforcing global smoothness while respecting local discontinuities in a learning framework striking a balance between complexity and accuracy.

Moreover, it demonstrates connections of the adaptive filtering approach with the self attention.

The experimentation on two challenging registration tasks cardiac and inter-subject abdominal registration demonstrate that AdaWarp outperforms existing methods in accuracy-efficiency and accuracy-smoothness tradeoffs.

**Weaknesses:**

Although I believe that the paper attempts to bridge a gap in the literature by incorporating a differentiable bilateral grid within a learning-based registration framework, I would like to point out several weaknesses and raise some questions regarding the experiments.

[A] I would like to invite the authors to elaborate on this statement regarding iterative optimization-based methods: “As a result, these approaches tend to be time-consuming and lack the ability to incorporate contextual information effectively.”

[B] “While high-dimensional filtering can project signals onto arbitrary spaces, we focus on extending by one additional dimension to account for the object boundary.”

What is the intuition behind this approach? Is only one additional dimension sufficient? I would like to invite the authors to further elaborate and explain their choice.

[C] The role of the guidance map generator component is unclear. Could the authors please explain why this component is used or needed?

[D] Could the authors clarify whether the same lambda values are used for all methods or if different values are applied? How were these values tuned? Were they also tuned for the baselines?

[E] The proposed method utilizes a diffeomorphic transformation model; however, it is not clear whether the baselines follow the same principle. Could the authors provide a table that explicitly lists the hyperparameters used by each of the baselines along with the transformation model?

[F] The authors chose different baselines for the two datasets, which is puzzling. What is the intuition behind this decision? Is there a reason why this approach was chosen?

[G] The paper presents t-tests for DICE scores but not for other metrics. Is there a reason for this choice? Could the authors extend their t-tests to cover HD95 as well?

[H] “Learning-based methods generally outperform traditional ones in registration accuracy, though with slightly higher SDlogJ.”

Do the authors have any intuition as to why this is the case? Normally, I would expect that iterative optimization methods achieve higher accuracy [1].

[1] Hansen, L. and Heinrich, M.P., 2021. Revisiting iterative highly efficient optimization schemes in medical image registration. In Medical Image Computing and Computer-Assisted Intervention–MICCAI 2021: 24th International Conference, Strasbourg, France, September 27–October 1, 2021, Proceedings, Part IV 24 (pp. 203-212). Springer International Publishing.

[I] For the abdominal dataset, the proposed method uses Convex Adam’s framework with the same segmentation model as a feature extractor. Is there any reason for this choice? Could the model be trained from scratch? Could the authors elaborate on the design choices, including why the architecture differs depending on the dataset?

[J] The code is not available. Are the authors planning to make their code publicly accessible?

[K] Due to the lack of ground truth, registration is evaluated quantitatively with surrogate measures. However, to ensure the registration’s success, it is common practice to inspect the resulting transformed images qualitatively as well. I would like to invite the authors to provide qualitative results for both datasets, as this would substantially strengthen their claims.

**Questions:**

I encourage the authors to consider addressing as many of the points highlighted in the weaknesses section as possible. Additionally, while the paper presents an intriguing and novel approach, the clarity and quality of the presentation could benefit from further refinement.

---

> ### Author Response · Authors · 2024-11-22
> **Part 1: Addressing [A], [B] and [C]**
>
> Dear Reviewer UMDT,
>
> We sincerely thank you for your valuable comments. Below, we address your points [A], [B], [C].
>
> [A] Reply: Thank you for pointing this out. We agree that the statement could be clarified. Specifically, traditional iterative methods can be categorized into two main types: continuous optimization-based and discrete optimization-based, each with distinct characteristics.
> - **Continuous optimization:** These methods typically linearize the dissimilarity term, requiring a series of small iterative updates. Each step accounts for only a minor deformation, which leads to slow convergence and difficulty in handling large deformations.
> - **Discrete optimization:** These methods are better suited for handling large deformations and require much fewer iterations to converge, making them favorable for datasets with large displacements. However, the primary drawback is their high memory consumption, which increases exponentially with the disparity volume size.
>
> By "the ability to incorporate contextual information," we refer to the advantage of learning-based methods in integrating label supervision, such as segmentation masks or keypoints. This integration is less straightforward in both continuous and discrete optimization-based approaches. While methods like ConvexAdam can incorporate segmentation masks generated by models like nnU-Net, such masks are unavailable without neural network-based learning. This limitation highlights the lack of contextual information typically inherent to iterative methods.
>
> We hope this explanation clarifies our statement.
>
> [B] Reply: We appreciate the reviewer’s attention to this detail. The proposed AdaWarp method addresses deformable image registration by incorporating the Piece-wise Smooth (P-S) Assumption. In this context, adding one additional range dimension is sufficient to achieve the desired effect. Since most medical image registration problems involve single-channel inputs, a single range dimension can effectively represent voxel-wise intensity differences. With a higher sampling rate $s_r$, this approach enhances edge distinction and ensures adequate sensitivity to object boundaries. Expanding to more dimensions could indeed be interesting, allowing the representation of contextual differences beyond raw intensity differences. Such an extension could generalize self-attention from adaptive Gaussian filtering to a broader learnable adaptive filtering framework. However, this direction is outside the scope of the current paper.
>
> [C] Reply: Thank you for highlighting this. A visual explanation of the bilateral grid process for a one-dimensional signal is shown in Fig. 2. In this example, the original signal is represented as $f: x \rightarrow \mathbb{R}$, where $f(x)$ corresponds to the image intensity at position $x$. When projecting to a higher-dimensional space (from 1D to 2D here) with an additional range dimension (image intensity), the image is represented as $f: (x, r) \rightarrow \mathbb{R}$, where $r$ denotes the intensity at position $x$. Here, $f(x, r)$ represents the corresponding intensity before blurring ($f(x, r) = r$). The guidance map serves as the additional coordinate to access elements in the bilateral grid. Instead of using raw image intensities directly as this coordinate, we employ a trainable guidance map generator. This design allows the network to adaptively learn a more effective coordinate representation, improving the flexibility and performance of the model.

---

> > ### Comment · Reviewer_UMDT · 2024-11-25
> > **Reply to Part 1.**
> >
> > I would like to thank the authors very much for engaging in the rebuttal and for their detailed responses and clarifications.
> >
> > [A] I am not sure if I am not understanding this correctly, but there were segmentation algorithms before deep learning, and at the same time, iterative optimization methods allowed multi-channel registration where one could incorporate contextual information (e.g. segmentation maps) along with the images. If misunderstanding is the case I think further clarification might be required in the paper to clarify this point.
> >
> > [B&C] I appreciate the clarification of these points. I strongly believe that these clarifications are missing from the paper and they make them stronger, clearer and more easy to follow. As a result I would like to invite the authors to include them in the paper.

---

> ### Author Response · Authors · 2024-11-22
> **Part 2: Addressing [F] and [H]**
>
> [F] Reply: Thank you for pointing this out. The choice of baselines differs due to the following considerations:
> 1. **Iterative methods (ANTs, Demons, Bspline):** These were included for the cardiac dataset but excluded for the abdomen dataset because our parameter tuning failed to yield reasonable performance on the latter.
> 2. **MemWarp and TextSCF:** MemWarp is included for the cardiac dataset as it was originally developed for this type, whereas TextSCF is included for the abdomen dataset for similar reasons.
> 3. **ConvexAdam and SAMConvex:** These methods are included for the abdomen dataset due to its characteristic large deformations. While they may also perform well on cardiac data, their primary design focuses on datasets with large deformations, making the baselines fair for the comparisons in this paper.
> 4. **DeBG:** As described in the original paper (lines 113–121 and 369–377), this method uses shuffled channels instead of real splatting to represent range dimensions, inaccurately preserving the image manifold in higher-dimensional space. As a result, this shuffling cannot handle the image pyramid-based approach used in the abdomen dataset, where each pyramid level shares the same feature map and cost volume across different scales.
>
> [H] Reply: Thank you for your question and the reference. We believe this question is crucial to the image registration community. The claim here is solely based on the results in Table 1, where learning-based methods generally outperform traditional iterative methods (ANTs, Demons, Bspline) in registration accuracy for the cardiac dataset used in this paper, though with slightly higher SDlogJ.
> However, this trend may not hold universally. The relative advantages of learning-based methods over iterative methods in purely unsupervised settings (without label supervision) remain an active research topic. Iterative methods, particularly discrete optimization-based ones, often achieve comparable accuracy to unsupervised learning methods on datasets with low-complexity deformations and higher accuracy on datasets with large deformations. This discrepancy stems from two main factors:
> 1. **Dissimilarity function:** Unsupervised learning methods share the same dissimilarity function as iterative methods, and amortized optimization provides no additional advantage for label matching performance in low-complexity scenarios. [2]
> 2. **Regularization:** In unsupervised learning, the burden of smoothness regularization is enforced entirely on the network weights, whereas iterative methods directly compute gradients backflow on the deformation field. This indirect optimization in unsupervised learning can lead to inferior performance compared to iterative methods, particularly on datasets with large deformations. Although some works [3] integrate smoothness regularization explicitly as part of the network, they still require segmentation masks for effective implementation.
>
> However, we have observed two cases where iterative methods may fall behind learning-based methods in image registration:
> 1. **Unsupervised settings with large-scale datasets:** In large-scale datasets with low-complexity deformations (e.g., the 4000-subject LUMIR dataset [4] from the Learn2Reg Challenge 2024 [5]), from our empirical experience, iterative methods, whether continuous, discrete, or instance optimization with neural networks, consistently underperform compared to pure learning-based methods, despite extensive parameter tuning. We attribute this gap to the regularization term. With sufficient data, while the dissimilarity metric offers no clear advantage for amortized optimization [2], the regularization term leverages the neural network’s ability to incorporate contextual information [3], enabling superior flow propagation and smoothness regularization.
> 2. **Semi-supervised settings with label supervision:** When label supervision (e.g., segmentation or keypoints) is introduced, learning-based methods outperform iterative methods by effectively utilizing surrounding contextual information from labels, resulting in higher registration accuracy. However, additional losses like segmentation loss may cause less smooth deformations at object boundaries, leading to implausible fields and higher SDlogJ values.
>
> [1] Hansen, L. and Heinrich, M.P., 2021. Revisiting iterative highly efficient optimization schemes in medical image registration. MICCAI 2021.
>
> [2] Jena, R., Sethi, D., Chaudhari, P. and Gee, J.C., 2024. Deep Learning in Medical Image Registration: Magic or Mirage?. arXiv preprint 2024.
>
> [3] Heinrich, M.P., 2019. Closing the gap between deep and conventional image registration using probabilistic dense displacement networks. MICCAI 2019
>
> [4] Liu, Y., Chen, J., Wei, S., Carass, A. and Prince, J., 2024. On finite difference jacobian computation in deformable image registration. IJCV 2024.
>
> [5] https://learn2reg.grand-challenge.org

---

> > ### Comment · Reviewer_UMDT · 2024-11-25
> > **Reply Part 2**
> >
> > [F] As one of the other reviewers pointed out, I believe that the results and discussion section is a bit unclear and difficult to understand. With these clarifications, I understand why they chose each method, but in my opinion, this should be included in the paper. However, I still find it quite peculiar that both datasets use different baselines. I would expect to see all the baselines for both datasets justifying why each did not work for each dataset.
> >
> > Moreover, I am aware that Abdominal datasets can be challenging, but I am not convinced that none of the ANTs, Demons, or B-spline can yield reasonable performance with careful hyperparameter tuning.
> >
> > The last remark for this point is that ANTs are a framework and not a transformation. One can, for example, use b-spline or SyN within ANTs framework.
> >
> > [H] Thank you for the clarifications on this point. I can now see the rational behind the writing.

---

> ### Author Response · Authors · 2024-11-25
> **Part 3: Addressing [D], [E] and [I]**
>
> [D&E] Reply: Thank you for pointing out the lack of descriptions for $\lambda$ values. We did not intentionally tune these hyperparameters but performed a grid search with $\lambda = 0.01, 0.1, 1.0$, and $5.0$. We found $\lambda = 0.01$ to be optimal for the ACDC dataset and $\lambda = 1.0$ for the abdomen dataset (note: the $\lambda = 5.0$ in the original manuscript was a typo). Most baseline methods use the same parameters and training settings as AdaWarp. Below, we provide more details.
>
> **ACDC Dataset:**
> All learning-based methods adopt the same hyperparameters as AdaWarp, with $\lambda = 0.01$, MSE as the dissimilarity loss, and scaling-and-squaring with 7 steps for the diffeomorphic transformation model. For Figure 4, while keeping other hyperparameters the same, we vary computational complexity by adjusting the starting channel count in FourierNet, LKU-Net, and Ada-Res, and by modifying the backbone of TransMorph (tiny, small, and normal).
>
> **Abdomen Dataset:**
> The abdomen dataset presents more challenges due to the large displacement problem. To clarify:
> - FourierNet, VoxelMorph, TransMorph, and Ada-Cost use local NCC as the dissimilarity loss with $\lambda = 1.0$. ConvexAdam and SAMConvex also use $\lambda = 1.0$, employing MIND and segmentation feature maps, respectively, to compute the dissimilarity. All these methods adopt scaling-and-squaring with 7 steps for the diffeomorphic transformation model.
> - For TextSCF, we follow its original implementation with $\lambda = 0.1$ and without the diffeomorphic transformation model. The $\lambda = 0.1$ version with integration is also presented in Figure 6.
> - Both LKUNet and LapIRN results in Table 2 use $\lambda = 1.0$ without the diffeomorphic transformation model. Additionally, we ran counterparts using scaling-and-squaring with 7 steps for the diffeomorphic transformation, and the results are as follows:
>
> **Abdomen Dataset:**
> | Model           | Type  | Dice (%) | HD95 (mm) ↓ | SDlogJ ↓ |
> |------------------|-------|----------|-------------|----------|
> | Initial          | -     | 30.86    | 29.77       | -        |
> | LKUNet           | L     | 52.78    | 20.56       | 0.98     |
> | LKUNet (diff)    | L     | 52.08    | 20.34       | 0.28     |
> | LapIRN           | L     | 54.55    | 20.52       | 1.73     |
> | LapIRN (diff)    | L     | 51.39    | 20.89       | 0.06     |
>
> Both methods show performance degradation in anatomical alignment, as measured by Dice, when required to produce smoother and more plausible deformation fields.
>
>
> [I] Reply: Thank you for pointing this out. This question is similar to the first one raised by Reviewer RMNZ, and we address the essentials here.
>
> **Why:** We do not intentionally use different architectures; the choice is application-driven. As also mentioned in our reply to Reviewer RMNZ, the core of this paper is the AdaWarp module, which generalizes to different architectures as needed.
> Initially, we developed Ada-Res for the ACDC dataset to achieve a reasonably good performance. However, applying Ada-Res to the abdomen dataset revealed suboptimal performance due to challenges like the aperture and large displacement problems. To address this, we tailored a solution for the abdomen dataset by incorporating image pyramids and discrete optimization, inspired by prior multi-scale approaches and ConvexAdam, leading to the development of Ada-Cost. Leveraging AdaWarp’s flexibility, we integrated these approaches effectively.
> While we hadn’t previously tested Ada-Cost on the ACDC dataset, we have now conducted these experiments as part of the rebuttal. The results are as follows:
>
> **Cardiac Dataset:**
> | Model            | Avg. (%) | RV (%) | LVM (%) | LVBP (%) | HD95 (mm) ↓ | SDlogJ ↓  |
> |------------------|----------|--------|---------|----------|-------------|-----------|
> | Initial          | 58.14    | 64.50  | 48.33   | 61.60    | 11.95       | -         |
> | Ada-Res (Ours)   | 79.20    | 78.14  | 76.31   | 83.15    | 8.33        | 0.050     |
> | Ada-Cost (Ours)  | 79.82    | 77.58  | 77.95   | 83.92    | 8.98        | 0.050     |
>
> This is interesting, as methods that perform well on ACDC may struggle on more challenging datasets like the abdomen dataset, whereas methods designed for challenging datasets like the abdomen dataset can be easily adapted to simpler datasets like ACDC. Note that for Ada-Cost results on ACDC, raw images were used as input, without feature maps from a segmentation network.

---

> > ### Comment · Reviewer_UMDT · 2024-11-25
> > **Reply Part3**
> >
> > [D&E] Thank you for the time and effort for these. I believe that all these should be either part of the paper or at least of the supplementary material.
> >
> > [I] Thank you very much for this clarification. Since Ada-Cost demonstrates superior performance (based on the table above), I am wondering what the purpose of Ada-Res is. Moreover, the need to adapt the method for the abdominal dataset raises the question if the method has to be adopted for every new dataset, especially for another modality or body part. I believe that this point needs to be examined again by the authors and to provide a better explanation in the paper.

---

> > > ### Comment · Reviewer_UMDT · 2024-11-25
> > > **Final remarks**
> > >
> > > As I said before, I want to thank the authors for engaging in the rebuttal. I believe that this is a novel work that brings merit in the field of registration. However, I believe that the paper is not very well organised, and as a result, this makes it unclear and difficult to follow. Many of the responses of the authors can be incorporated into the methods and results sections. Due to the amount of changes and additions, I believe that the paper would require another round of reviews to assess it.
> > >
> > > Last but not least, a very important point for me is, as I said in [K], the quantitative results (warped images, difference images, deformation field visualisations) as these allow all of us who work on registration to assess the quality along with the surrogate quantitative measures.
> > >
> > > Given all the above, without discouraging the authors (I still believe that the method is novel), I am not going to change my score with the hope that this paper will improve through this process and will be resubmitted later somewhere else.

---

> > > > ### Author Response · Authors · 2024-11-26
> > > > **Reply to 'Final remarks'**
> > > >
> > > > Dear Reviewer UMDT,
> > > >
> > > > Thank you for your critiques. We would like to provide a gentle reminder of the ICLR submission-review procedure and timeline, as mentioned in the email from the ICLR program chairs.
> > > >
> > > > **Resubmission:**
> > > > Unlike conferences like CVPR, ICLR allows revised submissions that address reviewer comments or include improvements. Therefore, we believe we can incorporate the feedback and reorganize the manuscript for greater clarity.
> > > >
> > > > **Timeline:**
> > > > We are still in the discussion phase and have not yet responded to all your comments or uploaded our revised manuscript. The timeline is as follows:
> > > > - **November 27th**: Last day to upload a revised PDF. After this date, only replies on the forum are allowed (no manuscript changes).
> > > > - **December 2nd**: Last day for reviewers to post messages to authors (six-day extension).
> > > > - **December 3rd**: Last day for authors to post messages on the forum (six-day extension).
> > > >
> > > > > *"Many of the responses of the authors can be incorporated into the methods and results sections. Due to the amount of changes and additions, I believe that the paper would require another round of reviews to assess it."*
> > > >
> > > > As we are working on the revised manuscript, we believe that if your concerns are addressed in the comments, the final version will reflect the necessary changes and additions.

---

> > > > ### Author Response · Authors · 2024-12-02
> > > >
> > > > Dear Reviewer UMDT,
> > > >
> > > > Thank you for your thoughtful critiques, which have greatly helped us improve the manuscript. As the deadline for the discussion phase approaches, we encourage you to provide any final feedback to address remaining concerns. After the deadline, we will no longer be able to respond, so your timely engagement is highly valued.
> > > >
> > > > To assist you in conducting **a subjective, transparent, and professional evaluation** of our work, we provide the following clarifications and reflections. Noting your **earlier unfamiliarity with the ICLR review procedure and timeline**, we hope this facilitates consensus and ensures the **real contributions and impact of our work are accurately reflected**.
> > > >
> > > > 1. **Addressed Concerns**:
> > > >    - We believe your prior comments **[A][B][C][D][E][G][H][I][J][K] have been addressed**. Some of these were explicitly acknowledged as resolved in your earlier responses, while others we believe have been addressed through the changes incorporated into the revised manuscript and summarized in the **Author Response Summary**.
> > > >    - If you feel any of these points remain unresolved, please let us know, and we will be happy to provide additional clarification.
> > > > 2. **Discussion on Baselines ([F])**:
> > > >    - While we acknowledge we may hold differing opinions on the choice of baselines for the two datasets, we believe this does not detract from the original contributions of the paper. We clarify the following points to ensure alignment:
> > > >      - **Iterative Methods**: Traditional iterative methods have been removed from the revised manuscript to avoid confusion. Their inclusion or exclusion does not affect the claims of the paper. Additionally, we note that such methods are significantly harder to tune compared to learning-based approaches, which we view as a key limitation of iterative methods.
> > > >      - **Dataset-Specific Methods**: Some learning-based methods used in ACDC and others in the abdomen dataset were chosen based on their design specificity and relevance to the respective tasks. Given the long training times for image registration frameworks, only methods designed for general use were evaluated across datasets. While including all methods for all datasets may aid in understanding, excluding certain task-specific methods does not impact the main claims or the comparison with state-of-the-art methods.
> > > >
> > > > Based on the above, we hope these clarifications address any remaining concerns. We kindly encourage you to reconsider and **ensure your final score reflects the contributions and broader impact of our work**. Your feedback has been invaluable, and we deeply appreciate your time and engagement.
> > > >
> > > > Sincerely,
> > > > The Authors

---

> > > > > ### Comment · Reviewer_UMDT · 2024-12-02
> > > > > **Final words and thoughts**
> > > > >
> > > > > Dear authors,
> > > > >
> > > > > First of all I would like to thank you for engaging with the rebuttal and for improving your manuscript taking into consideration all the reviewers comments and for providing quantitative results. As I said in my comments above I believe that this work will be interesting for the community.
> > > > > However, due to the amount of changes in the manuscript and the way that reviewers do/do not engage in the rebuttal I believe that the paper needs another round of reviews to ensure that the quality after the additions is met. Moreover, I still find the that the written quality of the manuscript should be improved, especially the results and discipline which I still find not optimally organized and discussed.
> > > > > More specifically, I am not persuaded about the choice of different baselines yet. In my opinion you could have chosen some common baselines that work for both and then support with strong arguments why you choose the different ones. Similarly, I am not sure I agree that the iterative methods are more difficult to tune and I certainly do not agree with the choice of removing them because they do not work. Although you hypothesise they probably wouldn’t affect the final result, this is not proven and in addition just removing them doesn’t show a good scientific practice in my opinion.
> > > > >
> > > > > Given all the above I am not raising my score. I believe this work has the potential to be a great paper for the registration community but I believe it needs to be worked a bit more in depth to improve and polish its experimentation.

---

> > > > > > ### Author Response · Authors · 2024-12-02
> > > > > >
> > > > > > Dear Reviewer UMDT,
> > > > > >
> > > > > > Thank you for your response and for acknowledging the interest this work holds for the registration community. However, as your comments continue to raise vague and subjective arguments, we feel it necessary to respond directly and clarify our position.
> > > > > >
> > > > > > > *"However, due to the amount of changes in the manuscript and the way that reviewers do/do not engage in the rebuttal I believe that the paper needs another round of reviews to ensure that the quality after the additions is met."*
> > > > > >
> > > > > > As you are so **rigorous**, could you specify **point by point** which changes require another round of reviews? It is unclear why you would resist reviewing a substantially improved manuscript or why these changes cannot be adequately assessed in this round.
> > > > > >
> > > > > > > *"More specifically, I am not persuaded about the choice of different baselines yet. In my opinion you could have chosen some common baselines that work for both and then support with strong arguments why you choose the different ones."*
> > > > > >
> > > > > > Didn’t we already include common baselines like VoxelMorph and newer ones like CorrMLP? If you disagree with our approach, please clarify your reasoning with examples of baselines you believe should have been included and explain how they would strengthen the study.
> > > > > >
> > > > > > > *"Similarly, I am not sure I agree that the iterative methods are more difficult to tune and I certainly do not agree with the choice of removing them because they do not work."*
> > > > > >
> > > > > > This statement appears vague and defensive, as though you may be a developer defending iterative methods rather than evaluating our paper. Nowhere in the manuscript do we claim that learning-based methods are universally better than traditional iterative methods. In fact, we explicitly state that in our response, learning-based methods often achieve similar or inferior performance compared to traditional methods in unsupervised settings.
> > > > > >
> > > > > > > *"Although you hypothesise they probably wouldn’t affect the final result, this is not proven and in addition just removing them doesn’t show a good scientific practice in my opinion."*
> > > > > >
> > > > > > What exactly do you believe needs to be proven? Our focus is on addressing gaps in learning-based methods and comparing them with other learning-based approaches. Including or excluding iterative methods does not change the core contributions of this paper, which are rigorously demonstrated. Simplifying the manuscript by removing confusing elements improves clarity and accessibility for readers, not the opposite.
> > > > > >
> > > > > > Finally, we want to emphasize that we are not here to argue about the merits of iterative versus learning-based methods. In fact, much of this work draws inspiration from efforts to bridge the gap between the two approaches such as [1][2][3]. Your continued focus on defending iterative methods rather than engaging with the actual contributions of the paper seems counterproductive to the review process.
> > > > > >
> > > > > > We encourage you to reflect on these points and reconsider your evaluation to ensure it is **transparent, fair, and objective**. The discussion period is ending soon, and we hope you can provide actionable feedback that reflects the contributions and impact of our work in a professional and constructive manner.
> > > > > >
> > > > > > Sincerely,
> > > > > > The Authors
> > > > > >
> > > > > > [1] Siebert, H., Großbröhmer, C., Hansen, L. and Heinrich, M.P., 2024. ConvexAdam: Self-Configuring Dual-Optimisation-Based 3D Multitask Medical Image Registration. IEEE Transactions on Medical Imaging.
> > > > > >
> > > > > > [2] Heinrich, M.P., 2019. Closing the gap between deep and conventional image registration using probabilistic dense displacement networks. In Medical Image Computing and Computer Assisted Intervention–MICCAI 2019: 22nd International Conference, Shenzhen, China, October 13–17, 2019, Proceedings, Part VI 22 (pp. 50-58). Springer International Publishing.
> > > > > >
> > > > > > [3] Heinrich, M.P., Papież, B.W., Schnabel, J.A. and Handels, H., 2014. Non-parametric discrete registration with convex optimisation. In Biomedical Image Registration: 6th International Workshop, WBIR 2014, London, UK, July 7-8, 2014. Proceedings 6 (pp. 51-61). Springer International Publishing.
> > > > > >
> > > > > > [4] Jena, R., Sethi, D., Chaudhari, P. and Gee, J.C., 2024. Deep Learning in Medical Image Registration: Magic or Mirage?. arXiv preprint arXiv:2408.05839.

---

> ### Author Response · Authors · 2024-11-29
> **Part 4: Addressing [G], [J], [K], and Further Clarification of [A]**
>
> [G] Reply:
> We have included t-tests for both Dice and HD95 metrics in the revised manuscript.
>
> [J] Reply:
> As noted in the first point of the **Author Response Summary**, we will release the code immediately upon the paper's acceptance.
>
> [K] Reply:
> For qualitative results, please refer to Figure 4 and Figure 10 in the revised manuscript. A brief summary of key observations is provided in the third point of the **Author Response Summary**.
>
> [A] Further Clarification:
> Thank you for your feedback on our response to your comment [A]. We have updated the manuscript to reflect the following discussion:
>
> - **Segmentation Accuracy:**
>   While there is ongoing debate on whether learning-based registration methods outperform traditional iterative methods, deep learning has undeniably dominated segmentation. Thus, existing iterative methods still rely on segmentation maps generated by deep learning, which provide superior contextual information compared to traditional segmentation algorithms.
>
> - **Amortized Optimization:**
>   Traditional iterative methods can incorporate contextual information (e.g., segmentation maps) but require instance-wise optimization, meaning new segmentation masks and energy function optimization are needed for every unseen image pair. In contrast, learning-based methods utilize amortized optimization, requiring only a single network trained on a cohort of image pairs, making them more efficient and scalable.

---

### Official Review · Reviewer_MgEu · 2024-11-01

**Soundness:** 3
**Presentation:** 3
**Contribution:** 3
**Rating:** 6
**Confidence:** 4

**Summary:**

This paper proposes a learning framework that improves the accuracy-efficiency trade-off in medical image registration by leveraging the piece-wise smooth prior. The proposed method was evaluated on two medical image datasets involving cardiac MRI and abdomen CT images. This method transforms the deformable registration problem into a keypoint detection task and shows potential for segmentation tasks.

**Strengths:**

The proposed method bridges the gap in the existing literature focusing on the balance between registration accuracy and computational efficiency, which is capable of enforcing global smoothness while respecting local discontinuities. This paper was well-written with very clear description on methodology.

**Weaknesses:**

1. The major concern is the research focus of this study, which might not be of sufficient significance in the field of medical image registration. After the introduction of deep learning-based registration methods, e.g., VoxelMorph, existing methods have been very fast in registration, allowing real-time registration using GPUs. Under this situation, only a few studies have specifically focused on improving efficiency, which suggests that this topic might not be the real problem in the community.
2. Another concern is the generalizability of the P-S assumption. In the study, this assumption was exemplified and evaluated with cardiac MRI and abdomen CT images, where there is no too many complex anatomical structures and local deformations. It’s important to evaluate the proposed method on the well-benchmarked brain MRI registration tasks, in which the P-S assumption may fail.

**Questions:**

1. In Figure 4 and Figure 5, why not include VoxelMorph into comparison? VoxelMorph is the most widely-benchmarked method and has high efficiency with low number of parameters.
2. There is a recent registration study in CVPR (CorrMLP, Meng et al. 2024), which is based on a totally conflicting motivation against this paper. CorrMLP attempted to capture long-range dependency among full-resolution image details in an efficient approach (using MLPs), while this paper suggests that only low-resolution features are sufficient. So, it’s interesting to compare with the CorrMLP: did the proposed method achieve similar registration accuracy while reducing much computational complexity?

---

> ### Author Response · Authors · 2024-11-24
> **Part 1: Addressing Concerns in Weakness Section**
>
> Dear Reviewer MgEu,
>
> We sincerely thank you for your valuable comments. Below, we address your concerns listed in the weakness section as [W1] and [W2], and answer your questions [Q1] and [Q2].
>
> [W1] Reply: We would like to emphasize that the focus of this study is not solely on improving registration efficiency, but on introducing a novel neural network architecture that achieves the best overall performance in image registration. This overall performance cannot be evaluated using a single surrogate metric, such as Dice. Instead, it requires a multidimensional comparison, taking into consideration of factors like accuracy-efficiency and accuracy-smoothness tradeoffs. We believe these aspects have been overlooked in previous studies, and we elaborate on them below.
> 1. **Accuracy-efficiency:** Evaluating accuracy or efficiency in isolation is insufficient. Our model demonstrates that with similar accuracy, it achieves lower computational complexity, and with similar computational complexity, it achieves higher accuracy. This balance is essential to claim better overall performance.
> 2. **Accuracy-smoothness:** A key advantage of learning-based methods is their ability to integrate label supervision. However, without adequately handling the smoothness of the deformation field, they can produce implausible or unrealistic deformations, even with high Dice scores. For example, using a pretrained nnU-Net for segmentation achieves over 90% Dice on ACDC and multi-organ abdomen segmentation tasks. If we then perform label matching [1] directly on the predicted masks, the Dice score improves further, but the resulting deformation field is often unrealistic due to the lack of smoothness and consideration of image textures.
>
> We hope this explanation addresses your concerns.
>
> [W2] Reply: Thank you for pointing this out. We would like to clarify that the cardiac MRI and abdomen CT datasets are indeed suitable testbeds for the proposed piece-wise smooth assumption. However:
>
> 1. **Brain MRI datasets and the piece-wise smooth assumption:** Brain MRI datasets do not violate the piece-wise smooth assumption. For adjacent regions with smooth transitions (e.g., left and right thalamus), these regions are treated as a single "piece" under the assumption, similar to existing models. For regions with clear boundaries (e.g., cerebral white matter and cortex), our model can perform better by explicitly handling such transitions.
> 2. **Relative difficulty of brain MRI datasets:** Brain MRI datasets are generally easier compared to cardiac and abdomen datasets. As noted in the second paragraph of the introduction in [2], abdominal scans are more complex than brain scans. Cardiac datasets exhibit local discontinuities and sliding motions, which are absent in brain MRI. Multi-organ abdomen datasets pose additional challenges:
>    - The **aperture problem** [3], arising in homogeneous or textureless regions, where the limited local evidence within a small window (defined by the network's *effective receptive field (ERF)* [4]) restricts accurate displacement estimation.
>    - The **large displacement problem**, where the displacement of a small structure between image pairs exceeds its own size, making accurate alignment more challenging.
>
> We hope this clarifies our rationale.
>
> [1] Durrleman, S., Prastawa, M., Charon, N., Korenberg, J.R., Joshi, S., Gerig, G. and Trouvé, A., 2014. Morphometry of anatomical shape complexes with dense deformations and sparse parameters. NeuroImage 2014.
>
> [2] Heinrich, M.P., 2019. Closing the gap between deep and conventional image registration using probabilistic dense displacement networks. MICCAI 2019.
>
> [3] Horn, B.K. and Schunck, B.G., 1981. Determining optical flow. Artificial intelligence 1981.
>
> [4] Luo, W., Li, Y., Urtasun, R. and Zemel, R., 2016. Understanding the effective receptive field in deep convolutional neural networks. NeurIPS 2016.

---

> ### Author Response · Authors · 2024-11-24
> **Part 2: Addressing Questions**
>
> [Q1] Reply: We agree that VoxelMorph is highly efficient with a low parameter count, and we did not exclude it intentionally from Figs. 5 and 6. The main reason for its exclusion is that VoxelMorph is not as competitive as other methods, and adding it would further crowd the figures. Here are its relevant statistics:
> 1. **Figure 5:** VoxelMorph would be positioned within the polygon formed by the lines of FourierNet and TransMorph, with Dice: 76.35%, Multi-Adds: 19.5 G, and Parameter Size: 0.32 MB.
> 2. **Figure 6:** VoxelMorph would appear near TransMorph, slightly to its upper left, with Dice: 47.05%, Multi-Adds: 73.30 G, and Parameter Size: 0.32 MB.
>
> [Q2] Reply: Thank you for mentioning CorrMLP. While CorrMLP has an interesting approach, we would like to clarify that **AdaWarp shares the same motivation as CorrMLP**. Although not explicitly stated in the manuscript, the low-resolution feature maps from deeper layers of neural networks inherently possess larger effective receptive fields (ERFs) than shallower layers.
> Please refer to the ERF visualization via [this link](https://ibb.co/BKtSJk2), also included in the revised manuscript appendix. Darker and more widely spread regions indicate larger ERFs. The details of ERF computation can be found in [1]. Key observations:
>    - **Leveraging Large Receptive Fields:** Subfigures (a), (b), and (c) illustrate feature maps from different encoder levels (L1: full resolution, L2: 1/2 downsampled, L3: 1/4 downsampled) from VoxelMorph. Deeper layers (e.g., L3) have larger ERFs, confirming that low-resolution features from deeper layers capture broader context. AdaWarp leverages the deepest encoder layer for the largest possible ERF. As shown in Table 4 (first row vs. last row), maintaining object boundaries (via AdaWarp) is essential for accuracy, as large ERFs alone are insufficient.
>    - **Effectiveness of AdaWarp Over Swin-Unet:** We compared ERF heatmaps of Swin-Unet and Ada-Swin (pre-softmax feature maps of models used in Table 4). Both share identical encoders, differing only in the decoder (Swin-Unet uses a U-Net structure, while Ada-Swin uses AdaWarp). Ada-Swin shows larger ERF regions and achieves 1.89% higher accuracy than Swin-Unet.
>
> As for the results of CorrMLP, we used the source code from their public repositories and ran experiments on the cardiac and abdominal datasets. The training settings were identical to those used for AdaWarp, including the same regularization strength and squaring-and-scaling integration with 7 steps. We have updated Tables 1 and 2 in the revised manuscript to reflect these results, and we briefly list them here for comparison.
>
> **Cardiac Dataset:**
> | Model          | Avg. (%) | RV (%) | LVM (%) | LVBP (%) | HD95 (mm) ↓ | SDlogJ ↓  |
> |----------------|----------|--------|---------|----------|-------------|-----------|
> | Initial        | 58.14    | 64.50  | 48.33   | 61.60    | 11.95       | -         |
> | CorrMLP        | 77.58    | 74.84  | 75.68   | 82.21    | 9.23        | 0.052     |
> | Ada-Res (Ours) | 79.20    | 78.14  | 76.31   | 83.15    | 8.33        | 0.050     |
>
>
> **Abdomen Dataset:**
> | Model             | Type  | Dice (%) | HD95 (mm) ↓ | SDlogJ ↓  |
> |-------------------|-------|----------|-------------|-----------|
> | Initial           | -     | 30.86    | 29.77       | -         |
> | CorrMLP           | L     | 56.58    | 20.40       | 0.16      |
> | Ada-Cost (Ours)   | L&D   | 62.74    | 15.03       | 0.12      |
>
> From the results, CorrMLP proves to be a highly competitive technique, outperforming all other baselines in registration accuracy while maintaining competitive smoothness of the deformation field. However, we observed a discrepancy between our reproduced results and those reported in Table 2 of the CorrMLP manuscript [2]. While we used the same data split for training and testing, they reported an Avg. Dice of 81.0%, whereas ours was 79.2%.
>
> Upon further investigation, we found this discrepancy may stem from differences in image preprocessing. CorrMLP resampled images to a voxel size of 1.5x1.5x3.15 mm³ and cropped to 128x128x32, while we resampled to 1.8x1.8x10 mm³ and cropped to 128x128x16. The ACDC dataset protocol [3] specifies slice thicknesses ranging from 5 mm to 10 mm and spatial voxel sizes between 1.34x1.34 and 1.68x1.68 mm².
>
> We are uncertain if CorrMLP's preprocessing aligns with standard practices, as upsampling from 5–10 mm to 3.15 mm in the axial direction introduces redundant information, potentially inflating the Dice score.
>
>
> [1] Luo, W., et al. Understanding the effective receptive field in deep convolutional neural networks. NeurIPS 2016.
>
> [2] Meng, M., et al. Correlation-aware Coarse-to-fine MLPs for Deformable Medical Image Registration. CVPR 2024.
>
> [3] Bernard, O., et al. Deep learning techniques for automatic MRI cardiac multi-structures segmentation and diagnosis: is the problem solved?. TMI 2018.

---

> ### Comment · Reviewer_MgEu · 2024-11-24
> **Final Review Comments**
>
> Thanks for the authors' effort in addressing my concerns. Overall, my concerns have been largely addressed, although I am still not convinced that brain MRI registration is an easier task than cardiac and abdomen registration. Anyway, the evaluation in the cardiac and abdomen datasets is fine for this study.  Nevertheless, after reading the comments from other reviewers, I think it's hard to increase my original score, so I decided to maintain it.

---

> > ### Author Response · Authors · 2024-12-02
> >
> > Dear Reviewer MgEu,
> >
> > Thank you for acknowledging that your concerns have been largely addressed. We sincerely appreciate your engagement and thoughtful feedback throughout the discussion.
> >
> > Regarding your statement, *“Nevertheless, after reading the comments from other reviewers, I think it's hard to increase my original score, so I decided to maintain it,”* we kindly ask for **clarification on how other reviewers' comments led to this conclusion**. We believe the score should primarily reflect your independent evaluation of the work, considering how well the concerns have been addressed and the broader contributions of the study.
> >
> > In particular, we encourage you to revisit the highlights of the broader impact and contributions of our work, as outlined in our latest comments. These include:
> > - Advancing the integration of physical priors (e.g., piece-wise smoothness) into neural networks, bridging classical image processing with modern architectures.
> > - Demonstrating strong performance on challenging datasets using a simple yet effective framework, providing a robust and accessible baseline for future work.
> > - Preliminary results that extend the method to tasks like keypoint-based lung motion estimation and segmentation, illustrating its versatility beyond registration.
> >
> > While the **discussion period is ending soon**, we value further dialogue to ensure that the contributions of this work are fairly assessed **in a professional, transparent, and independent manner**. We hope that this engagement reflects not only on the potential score but also on the broader impact this work may have on the image registration community and fundamental neural network development.
> >
> > Thank you again for your time and thoughtful input, and we look forward to any further comments you may have.
> >
> > Sincerely,
> > The Authors

---

> ### Author Response · Authors · 2024-12-01
> **Reply to Final Review Comments**
>
> Dear Reviewer MgEu,
>
> Thank you for your thoughtful review and for acknowledging that our responses have largely addressed your concerns. Since the primary issues have been resolved, we would like to highlight the broader impact and contributions of our work, which we hope might encourage you to **increase the score**.
>
> 1. **Interpretability and Simplicity**:
>    - Our method seamlessly incorporates a physical prior (piece-wise smoothness) into a neural network, enhancing interpretability while achieving strong performance.
>    - The architecture is **extremely simple**, making it easy to use and adapt for other researchers, while still setting a new baseline in learning-based registration.
>
> 2. **Connecting Attention Mechanism and Adaptive Filtering**:
>    - By introducing learnable adaptive filtering, we have drawn initial connections between self-attention and learnable adaptive filtering, as well as between gated attention and learnable adaptive filtering. This potentially generalizes modern neural network architectures and positions learnable adaptive filtering as **a foundational block for the next generation of neural networks**.
>
> 3. **Connecting Traditional Algorithms and Modern Neural Networks**:
>    - In addition, our method bridges the gap between traditional image processing methods, such as bilateral filtering, and modern neural network mechanisms. This fusion not only enriches the medical imaging field but also offers a pathway for integrating classical techniques into contemporary deep learning frameworks.
>
> 4. **Broader Applicability**:
>    - Beyond the cardiac and abdomen datasets, we have demonstrated AdaWarp’s versatility with preliminary results on keypoint-based lung motion estimation and medical image segmentation. These results showcase its potential to address diverse and critical tasks for the community.
>
> Given that our responses have addressed your concerns and our work contributes to the community through its interpretability, simplicity, and broader impact, we kindly request that you reconsider and further increase your score. We believe our study offers meaningful insights for researchers and practitioners alike.
>
> Thank you again for your engagement and valuable feedback.

---

### Official Review · Reviewer_TEi6 · 2024-11-05

**Soundness:** 3
**Presentation:** 2
**Contribution:** 2
**Rating:** 5
**Confidence:** 4

**Summary:**

This paper leverages prior knowledge observed in medical images to introduce the Piece-wise Smooth (P-S) Assumption as a basis for addressing medical image registration tasks. Specifically, the authors propose AdaWarp, a warping method that utilizes learnable adaptive filtering to register medical scans in line with the P-S assumption. By employing a low-resolution latent representation along with a differentiable bilateral grid, the method achieves a better balance between accuracy and efficiency. Experiments conducted on two registration datasets validate the effectiveness of the proposed approach.

**Strengths:**

1. The motivation behind this paper is reasonable. By analyzing daily CT and MRI scans in the cardiac and abdominal regions, the authors observed two consistent patterns across certain subjects, leading to the formulation of the Piece-wise Smooth (P-S) Assumption. This assumption leverages physical priors from observed medical image patterns, which is both innovative and plausible, enhancing neural network-based registration tasks by grounding them in realistic assumptions about medical image structures.
2. The paper provides thorough comparative experiments. The authors test AdaWarp on two registration datasets spanning different modalities and input constraints, which demonstrates robustness and broad applicability.

**Weaknesses:**

1. The novelty of this paper does not seem particularly strong. While the method leverages an encoder to extract a latent representation that approximates the deformation field at a low resolution, this approach mainly contributes to the model's efficiency but is not unique. The use of latent feature representations for similar tasks has already become common in the field.
2.  The core of AdaWarp is a differentiable bilateral grid,  which naturally incorporates the P-S prior.  In implementation, the guidance map aids in processes like splatting, blurring, and slicing. This incremental modification lacks sufficient novelty.

**Questions:**

See the above strengths and weaknesses.

**Details Of Ethics Concerns:**

N.A

---

> ### Author Response · Authors · 2024-11-29
> **Reply to Weakness Section**
>
> Dear Reviewer TEi6,
>
> Thank you for your critiques regarding the novelty of the manuscript. Please refer to the revised manuscript and the 2nd point in the **Author Response Summary** for detailed explanations of our claims of technical novelty. Specifically addressing your concerns:
>
> 1. We agree that latent feature representations are a common technique used in most neural networks, and we did not claim this as novel. However, the major novelty of our work lies in addressing a significant gap in the registration literature by seamlessly incorporating the physical prior, i.e., the piece-wise smooth assumption, into neural networks in an end-to-end trainable manner. To the best of our knowledge, no existing literature has achieved this.
>
> 2. We understand that simple incremental modifications may lack sufficient novelty. While a differentiable bilateral grid may appear straightforward at first glance, there is no fully functional and trainable bilateral grid for learnable adaptive filtering in either the registration or natural image processing literature. As discussed in the revised manuscript and the 2nd point in the **Author Response Summary**, the closest methods, such as deep bilateral grids [1][2], use channel shuffling to inadequately represent the range dimension, resulting in suboptimal performance. Our AdaWarp addresses this limitation and holds broader implications for future applications in image processing.
>
>
> [1] Gharbi, M., Chen, J., Barron, J.T., Hasinoff, S.W. and Durand, F., 2017. Deep bilateral learning for real-time image enhancement. ACM Transactions on Graphics (TOG), 36(4), pp.1-12.
>
> [2] Xu, B., Xu, Y., Yang, X., Jia, W. and Guo, Y., 2021. Bilateral grid learning for stereo matching networks. In Proceedings of the IEEE/CVF Conference on Computer Vision and Pattern Recognition (pp. 12497-12506).

---

> ### Author Response · Authors · 2024-12-02
>
> Dear Reviewer TEi6,
>
> We hope this message finds you well. As the deadline for the discussion period approaches, we would like to kindly remind you that we **have not yet received any follow-up feedback from you**. To make progress, we would appreciate it if you could clarify **a few points** from your previous review.
>
> >*"While the method leverages an encoder to extract a latent representation that approximates the deformation field at a low resolution, this approach mainly contributes to the model's efficiency but is not unique."*
>
> To make this critique more actionable, could you **suggest specific approaches that improve efficiency in image registration** and provide a comparison of their **pros and cons** relative to the proposed method? This would help us better contextualize and address your concern.
>
> >*"The use of latent feature representations for similar tasks has already become common in the field."*
>
> We would like to clarify that our paper **does not claim the use of latent feature representations as a novelty or contribution**. This point is merely a common component in modern neural network architectures.
>
> >*"The core of AdaWarp is a differentiable bilateral grid, which naturally incorporates the P-S prior. In implementation, the guidance map aids in processes like splatting, blurring, and slicing. This incremental modification lacks sufficient novelty."*
>
> Could you elaborate on why this is considered an incremental modification? To substantiate your claim, we kindly ask you to address the following questions:
> - Are there any existing methods that perform the same process as a fully differentiable bilateral grid? If so, can you **list them and show how** they compare to ours in terms of **novelty and functionality**?
> - **Which part** specifically do you consider incremental? Could you **point out the subtle modifications** we have made that lead to this characterization?
>
> Your insights would be invaluable in helping us refine our work and better address the concerns raised. We sincerely appreciate your time and look forward to your response.
>
> Best,
> The Authors

---

> > ### Comment · Reviewer_TEi6 · 2024-12-02
> > **thanks for reply**
> >
> > Thank you, authors, for your active engagement and thorough responses during the rebuttal phase. After carefully reading all the reviewers' comments alongside your replies, I decided to maintain my initial score finally.

---

> > > ### Author Response · Authors · 2024-12-02
> > >
> > > Dear Reviewer TEi6,
> > >
> > > Thank you for your engagement (if any) during the rebuttal phase. However, your decision to maintain the initial score, despite our detailed responses and revisions, **lacks constructive critique or justification**. Simply labeling our work as "incremental" without evidence or actionable feedback is **unhelpful and falls short of the professionalism expected in an ICLR review process**. We hope you reflect on this, as **fairness and transparency are principles** that the academic community ultimately upholds.
> > >
> > > Sincerely,
> > > The Authors

---

### Author Response · Authors · 2024-11-29
**Author Response Summary**

Dear Reviewers,

We sincerely thank you all for your valuable comments, which have greatly helped us improve the clarity and presentation of our manuscript. We have uploaded a revised version incorporating most of your comments and suggestions. Below, we summarize the key clarifications and major changes made to the manuscript for your reference.

1. **Code Release**:
   In response to comment [J] from Reviewer **UMDT** and for the benefit of potential readers, we confirm that the source code will be released immediately upon acceptance of the paper.

2. **Novelty**:
   We sincerely thank Reviewer **UMDT** and **RMNZ** for acknowledging the novelty and potential future impact of this work on the image registration community. We also appreciate Reviewer **MgEu** for recognizing that we have largely addressed their concerns regarding the research focus and the utility of the piece-wise smooth assumption. For Reviewer **TEi6** and potential future readers, we summarize the key *technical innovations* of this manuscript below:
   - **Physical Prior**: Current learning-based registration frameworks lack an end-to-end learnable approach to integrate the physical prior, i.e., the piece-wise smooth assumption, into neural networks, resulting in suboptimal performance. While this assumption is well-suited to medical images, none of existing methods have adequately addressed it.
   - **Differentiable Bilateral Grid**: To our knowledge, no prior work has proposed an end-to-end trainable bilateral grid with learnable adaptive filtering. The closest existing method, DeBG (discussed in the manuscript), employs channel shuffling instead of real splatting for the range dimension, leading to suboptimal results.
   - **Registration Performance**: By integrating AdaWarp, we achieve state-of-the-art registration performance using a simple network architecture with only a handful of convolutional layers. This surpasses existing learning-based models and could serve as a new milestone in image registration, providing a strong baseline with significant potential for further improvement.

3. **Qualitative Results**:
   In response to comment [K] from Reviewer **UMDT** and comments [W2] and [Q3] from Reviewer **RMNZ**, as well as for the benefit of other reviewers, we have included qualitative results for both datasets in the revised manuscript. Please refer to Figure 4 for the abdomen dataset and Figure 10 (in the appendix) for the cardiac dataset. Below, we briefly summarize key observations:
   - **Summary**: Our method achieves a piece-wise smooth displacement field, ensuring smooth displacements within regions while allowing differences between regions based on local motion.
   - **Cardiac Dataset**: AdaWarp performs better at the boundary between the right ventricle and left ventricle myocardium, where other methods show disorganized displacements. Additionally, unlike other methods that display two shrinking centers during registration from the end-diastole to end-systole phase, AdaWarp more realistically produces a single center.
   - **Abdomen Dataset**: AdaWarp effectively handles large deformations and captures local discontinuities. The sharp changes near boundaries (e.g., left/right kidneys, liver, and between the body and background) demonstrate its ability to incorporate the piece-wise smooth prior.

4. **Further Clarification on $\lambda$**:
   In the first version of the manuscript, we reported that Ada-Cost achieved a Dice score of 62.74%, which was based on $\lambda=5.0$. In accordance with other baseline methods, we have updated the table with results for $\lambda=1.0$. For tables in both datasets, all learning-based methods (except TextSCF) were trained under the same configuration, using scaling-and-squaring with 7 steps for diffeomorphic transformations.

5. **Further Clarification on Baselines**:
   Since AdaWarp focuses on addressing challenges in learning-based registration frameworks, we have removed traditional iterative methods from the revised manuscript to avoid confusion. Their inclusion or exclusion does not impact the claims of the paper. However, we note that traditional iterative methods are significantly harder to tune compared to learning-based approaches, which we consider a key limitation of such methods.

---

### Meta-Review · Area_Chair_Ai5N · 2024-12-19

**Metareview:**

This paper presents a method for deformable medical image registration. The main contribution of the authors is to introduce P-S assumption to enforce global smoothness while respecting local discontinuities. While this assumption is newly introduced to deep learning based registration, it is not a new concept and has been used in traditional registration as well as other deep learning tasks. Therefore, I do agree with reviewers that the novelty of this work is limited. The reviewers also have concern on the presentation of the paper.  As pointed out by reviewer UMDT, "Many of the responses of the authors can be incorporated into the methods and results sections. Due to the amount of changes and additions, I believe that the paper would require another round of reviews to assess it."
I also realized that this paper use cardiac and abdomen datasets in their experiments. However, a large amount of deformable registration work deals with brain datasets (which are publicly available). I would suggest the authors to include these datasets to improve the generality of the work.
I have also considered the concerns from the authors regarding some of the reviewer comments. I agree that some reviewers may have high standard/requirements for papers. However, the rebuttal is mainly used to clarify the potential misunderstanding of the papers, instead of getting reviewers to help revise or even rewritten the papers. Therefore, I believe it is the responsibilities of the authors to make the paper clear and convincible to the audience.

Given these weaknesses and the fact that the overall scores of the paper are not high, I cannot recommend to accept this paper.

**Additional Comments On Reviewer Discussion:**

The reviewers raised some issues of the paper which they feel the required changes is too much for a simple revision. Therefore, they believed that a resubmission to next venue with another round of review is more appropriate. The reviewers also felt that the contribution is not very strong, which is a subjective evaluation though.

---

### Decision · Program_Chairs · 2025-01-22

Reject